# Convergent Reinforcement Learning with Function Approximation: A Bilevel Optimization Perspective

## Abstract

We study reinforcement learning algorithms with nonlinear function approximation in the online setting. By formulating both the problems of value function estimation and policy learning as bilevel optimization problems, we propose online Q-learning and actor-critic algorithms for these two problems respectively. Our algorithms are gradient-based methods and thus are computationally efficient. Moreover, by approximating the iterates using differential equations, we establish convergence guarantees for the proposed algorithms. Thorough numerical experiments are conducted to back up our theory.

## 1 Introduction

In reinforcement learning (Sutton & Barto, 1998), the agent aims to make optimal decisions by interacting with the environment and learning from the experiences, where the environment is modeled as a Markov Decision Process (MDP). With the recent advancement of deep learning, reinforcement learning has achieved extraordinary empirical success in solving complicated decision making problems, such as the game of Go (Silver et al., 2016; 2017), navigation (Banino et al., 2018), and dialogue systems (Li et al., 2016).

Despite its great empirical success, there exists a gap between the theory and practice of reinforcement learning. Specifically, in terms of theory, most existing works focus on either the tabular case or the case with linear function approximation. In the former case, both the state and action spaces are finite, while in the latter the value function is assumed to be linear in a given feature mapping. Using tools for convex optimization and linear regression, the statistical and computational properties of the reinforcement learning algorithms are well-understood under these restrictive settings. Moreover, when it comes to nonlinear function approximation, theoretical analysis becomes intractable as it involves solving a highly nonconvex statistical optimization problems. Moreover, it is shown in Tsitsiklis & Van Roy (1997) that simple method such as TD(0) might fail to converge when nonlinear value functions are applied. However, using deep learning techniques, methods such as deep Q-network (DQN) (Mnih et al., 2015) and asynchronous advantage actor-critic (A3C) (Mnih et al., 2016) become baseline algorithms for artificial intelligence for practical applications.

To bridge such a gap in DRL, we make the first attempt to study the convergence of online reinforcement learning algorithms with nonlinear function approximation in general. In particular, we propose online Q-learning (Watkins & Dayan, 1992) and actor-critic algorithms with two-timescale updates Konda & Tsitsiklis (2000), i.e., the iterative algorithm consists of two parameters that updates with different learning rates. Although the statistical and computational properties of these methods with linear function approximation are well-studied, convergence of reinforcement learning methods with nonlinear function approximation is fundamentally more challenging due to the lack of convexity. In this case, we can only expect convergence to local minima or stationary points.

Furthermore, we bring both the problems of value function and policy learning under the unified framework of bilevel optimization, which motivates two-timescale updating rules in our algorithms.

Specifically, bilevel optimization problems consist of two subproblems that are intertwined with each other. Such phenomenon appears in both Q-learning and actor-critic algorithms. For Q-learning with function approximation, the updates of the target function (also known as target network for DQN) and the value function are coupled together; the same is the updates of the policy and value function in the actor-critic algorithm. Since the two variables in bilevel optimization have different roles, one servers as the constraint of the other, two-timescale updates appears naturally for these problems. In particular, the parameter of the lower optimization problem needs to be updated using a larger stepsize. In addition, as a byproduct, from the view of bilevel optimization, the target network in DQN appears to be the parameter of the upper level optimization, which justifies the trick of using a target network in DQN.

The analysis of our algorithms utilizes stochastic approximation (Borkar, 2008; Kushner & Yin, 2003), which approximates the asymptotic behavior of the iterates using ordinary differential equations. Specifically, under certain assumptions, we show that the online Q-learning algorithm converges almost surely to a local minimizer of the mean squared Bellman error. In addition, for the actor-critic algorithm, we show that the limit points of the actor and critic updates can be characterized by a set of equations.

Our contributions are three-fold. First, we unify the problems of value function estimation and policy learning using the framework of bilevel optimization, which leads to the online Q-learning and actor-critic algorithms using two-timescale updates. Second, using stochastic approximation, the convergence of the proposed algorithms are established with nonlinear function approximation. Third, as a byproduct, the formulation of Q-learning via bilevel optimization justifies the techniques of target network used in DQN, which can be viewed as the parameter of the upper level optimization subproblem.

**Related Work.** Our work is closely related to the literature on the convergence of online reinforcement learning algorithms with function approximation. For policy evaluation, most existing works focus on algorithms with linear function approximation. Tsitsiklis & Van Roy (1997) study the convergence of the on-policy TD($\lambda$) algorithm based on temporal-difference (TD) error. To handle off-policy sampling, Maei et al. (2010); Sutton et al. (2009; 2016); Yu (2015); Hallak & Mannor (2017) propose various TD-learning methods with convergence guarantees. Utilizing two-timescale stochastic approximation in Borkar (2008) and strong convexity, they establish global convergence results for the proposed methods. The finite-sample analysis of these methods are recently established in Dalal et al. (2017b;a). Moreover, using a duality formulation, Liu et al. (2015); Du et al. (2017) study the finite-sample performance of TD-learning algorithms with primal-dual updates. As for nonlinear function approximation, to the best of our knowledge, the only convergent algorithm is the nonlinear-GTD algorithm proposed in Bhatnagar et al. (2009a), which is more pertinent to our results. Their analysis also depends on two-timescale stochastic approximation, where the variable in the faster timescale essentially solves a linear equation, thus converges to the global optimum. However, in our problem, both the two variables solves nonconvex problems, which adds more challenge in the analysis. In addition, we mainly consider the problem of Q-learning and actor-critic, whereas their results only focus on policy evaluation. Moreover, their algorithm involves the Hessian of the value function and their theory requires the value function to be three times differentiable, whereas our algorithm only requires the gradient of the value function with less stringent assumptions. Thus, our results are not directly comparable with this work.

Furthermore, for Q-learning and the actor-critic algorithm, existing results all consider linear function approximation. For Q-learning, Melo et al. (2008); Prashanth et al. (2014) study the convergence using stochastic approximation. In addition, the actor-critic algorithm is introduced in Sutton et al. (2000); Konda & Tsitsiklis (2000), which also study its convergence with linear function approximation. Later, Kakade (2002); Peters & Schaal (2008) propose the natural actor-critic algorithm which updates the policy function using natural gradient descent (Amari, 1998). Convergence of actor-critic and natural actor-critic algorithms with linear function approximation are studied in Bhatnagar et al. (2009b; 2008); Castro & Meir (2010); Bhatnagar (2010); Maei (2018).

## 2 NOTATION AND BACKGROUND

In this section, we first lay out the notation that will be followed throughout this paper and then introduce some background knowledge on reinforcement learning.

We stick to the following notation. For a measurable space with domain $\mathcal{S}$, we denote by $B(\mathcal{S})$ the set of bounded measurable functions on $\mathcal{S}$. Let $\text{Lip}(\mathcal{S}, L)$ be the set of Lipschitz continuous functions on $\mathcal{S}$ with Lipschitz constant $L$. Let $\mathcal{P}(\mathcal{S})$ be the set of all probability measures over $\mathcal{S}$. For any $\nu \in \mathcal{M}(\mathcal{S})$ and any measurable function $f \colon \mathcal{S} \to \mathbb{R}$, we denote by $\|f\|_{\nu,p}$ the $\ell_p$-norm of $f$ with respect to measure $\nu$. For an integer $k$, we use $\Delta_k$ to denote the probability simplex in $\mathbb{R}^k$, and let $[k] = \{1, 2, ..., k\}$.

In reinforcement learning, the problem is modeled by a Markov decision process. A discounted MDP is defined by a tuple $(\mathcal{S}, \mathcal{A}, P, R, p_0, \gamma)$, where $\mathcal{S}$ is the set of states, $\mathcal{A}$ is the set of all possible actions, $P \colon \mathcal{S} \times \mathcal{A} \to \mathcal{P}(\mathcal{S})$ is the Markov transition kernel, $R \colon \mathcal{S} \times \mathcal{A} \to \mathbb{R}$ is the reward function, $p_0 \in \mathcal{P}(\mathcal{S})$ is the distribution of the initial state $s_0$, and $\gamma \in (0, 1)$ is the discount factor. At any state $s \in \mathcal{S}$, suppose action $a \in \mathcal{A}$ is executed, we use $P(\cdot \,|\, s, a) \in \mathcal{P}(\mathcal{S})$ to denote the probability distribution of the next state, and let $R(s, a)$ be the the immediate reward. For regularity, we assume that the rewards are uniformly bounded by $R_{\max}$. By taking action $a_t$ at state $s_t$ for each time step $t \geq 0$, the agent receives the discounted cumulative reward $R = \sum_{t \geq 0} \gamma^t \cdot R(s_t, a_t)$.

Moreover, a policy $\pi \colon \mathcal{S} \to \mathcal{P}(\mathcal{A})$ is a action selection rule. Specifically, $\pi(a \,|\, s)$ is the probability of selecting action $a$ at state $s$. We define the state value function and the action value function of policy $\pi$ as $V^\pi(s) = \mathbb{E}_\pi(R \,|\, s_0 = s, \pi)$ and $Q^\pi(s, a) = \mathbb{E}_\pi(R \,|\, s_0 = s, a_0 = a)$, respectively. Here we use $\mathbb{E}_\pi$ to indicate that the actions $\{a_t\}_{t \geq 1}$ follow policy $\pi$. We define Bellman operator of $\mathcal{T}^\pi \colon \mathcal{B}(\mathcal{S}) \to \mathcal{B}(\mathcal{S})$ by $\mathcal{T}^\pi V(s) = \mathbb{E}[r(s_t, a_t) + \gamma \cdot V(s_{t+1}) \,|\, s_t = s]$, where $a_t \sim \pi(\cdot \,|\, s_t)$. By definition, we have $V^\pi = \mathcal{T}^\pi V^\pi$ for any policy $\pi$. Since $\mathcal{T}^*$ is $\gamma$-contractive, $V^\pi$ is the unique fixed point of $\mathcal{T}^\pi$. The problem of estimating $V^\pi$ for a fixed policy $\pi$ is called policy evaluation.

Furthermore, in reinforcement learning, the goal is to obtain the policy that yields the maximal cumulative reward in expectation. To this end, we define the optimal value function $Q^* \colon \mathcal{S} \times \mathcal{A} \to \mathbb{R}$ by $Q^*(s, a) = \sup_\pi Q^\pi(s, a), \forall (s, a) \in \mathcal{S} \times \mathcal{A}$, where the supremum is taken over all possible policies. Then it is known that the greedy policy with respect to $Q^*$ is the optimal policy. To estimate $Q^*$ with accuracy, most existing methods utilize the fact that $Q^*$ is the unique fixed point of the Bellman optimality operator $\mathcal{T}^*$, which is defined by

$$(\mathcal{T}^* Q)(s, a) = r(s, a) + \gamma \cdot \mathbb{E}\big[\max_{a \in \mathcal{A}} Q(s_{t+1}, a) \,\big|\, s_t = s, a_t = a\big]. \tag{2.1}$$

The most well-known method for estimating $Q^*$ is the Q-learning algorithm (Watkins & Dayan, 1992), based upon a large family of algorithms are derived.

In addition to methods that focus on estimating the value functions, another class of algorithms in reinforcement learning directly search for a good policy over a parametrized policy class $\{\pi_\omega \colon \omega \in \mathbb{R}^p\}$ directly, where $\omega$ is the parameter. Specifically, the goal is to maximize $J(\omega) = \mathbb{E}_{\pi_\omega}(R) = \mathbb{E}_{s_0 \sim p_0}[V^{\pi_\omega}(s_0)]$. The basis of these methods are provided by the policy gradient theorem (Baxter & Bartlett, 2000), which states that

$$(1 - \gamma) \cdot \nabla_\omega J(\omega) = \mathbb{E}_{s \sim \rho^\omega, a \sim \pi_\omega(\cdot \,|\, s)}[\nabla_\omega \log \pi_\omega(a \,|\, s) \cdot Q^{\pi_\omega}(s, a)]$$
$$= \mathbb{E}_{s \sim \rho^\theta, a \sim \pi_\omega(\cdot \,|\, s)}\big\{\nabla_\omega \log \pi_\omega(a \,|\, s) \cdot [R(s, a) + \gamma \cdot V^{\pi_\omega}(s') - V^{\pi_\omega}(s)]\big\}, \tag{2.2}$$

where $s' \sim P(\cdot \,|\, s, a)$ is the next state of the MDP given $(s, a) \in \mathcal{S} \times \mathcal{A}$, and $\rho^\omega \in \mathcal{P}(\mathcal{S})$ is the state occupancy measure of $\pi_\omega$, i.e., $\rho^\omega(s) = (1 - \gamma) \cdot \sum_{t \geq 0} \gamma^t \cdot \mathbb{P}_{\pi_\omega}(s_t = s)$. We note that (2.2) brings the value function and the policy together. By updating $\theta$ using (2.2) with the value function $Q^{\pi_\omega}$ or $V^{\pi_\omega}$ estimated using temporal difference learning methods, we obtain the actor-critic algorithms (Konda & Tsitsiklis, 2000).

## 3    REINFORCEMENT LEARNING AS BILEVEL OPTIMIZATION

As we introduced in the previous section, estimation of the value functions and policy learning are crucial problems of reinforcement learning. In this section, we show that both these two problems can be cast into bilevel optimization. From this view, we establish the $Q$-learning and actor-critic algorithms with two-timescale updates.

In the unconstrained form, bilevel optimization is formulated as an optimization problem that contain another optimization problem as a constraint. Specifically, it can be written as

$$x^* = \underset{x \in \mathcal{X}}{\operatorname{argmin}} \, F[x, y^*(x)], \qquad \text{where} \;\; y^*(x) = \underset{y \in \mathcal{Y}}{\operatorname{argmin}} \, G(x, y), \tag{3.1}$$

where $F, G \colon \mathcal{X} \times \mathcal{Y} \to \mathbb{R}$ are two differentiable functions. Here the challenges in (3.1) are threefold. First, to evaluate the objective function $F[x, y^*(x)]$, one needs to achieve the global minimizer of $G(x, y)$ for each $x$, which might not be feasible when $G(x, y)$ is not a convex function of $y$. Second, even if we are able to obtain $y^*(x)$ and evaluate $F[x, y^*(x)]$ for any $x \in \mathcal{X}$, since it is impossible to compute the gradient of $y^*(\cdot)$, minimize $F[x, y^*(x)]$ via gradient-based methods is impossible. In practice, we can only hope for obtaining $(x^\natural, y^\natural) \in \mathcal{X} \times \mathcal{Y}$ satisfying

$$y^\natural = y^*(x^\natural) \qquad \text{and} \qquad x^\natural = \underset{x \in \mathcal{X}}{\operatorname{argmin}} \, F(x, y)\,\big|_{\,y = y^\natural}. \tag{3.2}$$

Third, in machine learning applications, usually both $F$ and $G$ in (3.1) are computed using large-scale datasets. In this situation, it can be highly costly to solve the lower level optimization problem to the exact minimizer. Thus, efficient algorithms have to be robust to the error incurred in solving for $y^*(\cdot)$.

An efficient algorithm that achieves the condition in (3.2) is two-timescale gradient descent, where we simultaneously perform gradient update with respect to both $x$ and $y$. Specifically, we have

$$x_{t+1} \leftarrow x_t - \alpha_t \cdot \nabla_x F(x_t, y_t), \qquad y_{t+1} \leftarrow y_t - \beta_t \cdot \nabla_y G(x_t, y_t), \tag{3.3}$$

where the stepsizes $\{\alpha_t, \beta_t\}_{t \geq 0}$ satisfy the following two-timescale assumption.

**Assumption 3.1.** We assume that the stepsizes $\{\alpha_t, \beta_t\}_{t \geq 0}$ satisfy

$$\sum_{t \geq 0} \alpha_t = \sum_{t \geq 0} \beta_t = \infty, \qquad \sum_{t \geq 0} \alpha_t^2 + \beta_t^2 < \infty, \qquad \lim_{t \to \infty} \alpha_t / \beta_t = 0. \tag{3.4}$$

That is, in addition to the standard Robbins-Monro condition (Robbins et al., 1951), we have $\alpha_t / \beta_t \to 0$, which indicates that $\{y_t\}_{t \geq 0}$ updates in a faster pace than $\{x_t\}_{t \geq 0}$. This condition gives the name of two-timescale update, and plays a pivotal role in the theoretical analysis.

More rigorously, measured in the faster timescale using $\{\beta_t\}_{t \geq 0}$, the updates in (3.3) converge to the asymptotically stable equilibria of the ODE system $\{\dot{x} = 0, \dot{y} = -\nabla_y G(x, y)\}$. Thus, $\{x_t, y_t\}_{t \geq 0}$ converges to some $(x^\natural, y^\natural)$ where $y^\natural$ is a local minimizer of $G(x^\natural, \cdot)$. When $G(x^\natural, \cdot)$ has a unique minimizer, we have $y^\natural = y^*(x^\natural)$. Thus, under the faster timescale, the updates in (3.3) essentially fix the $x$ variable and let $\{y_t\}_{t \geq 0}$ converge to $y^*(x)$ using gradient descent.

Furthermore, to determine the convergence of $\{x_t\}_{t \geq 0}$, we need to look into the slower timescale. Since $\{y_t\}_{t \geq 0}$ updates in a faster speed under Assumption 3.1, we could safely assume that $\{y_t\}_{t \geq 0}$ already converge to $y^*(x)$ for some $x \in \mathcal{X}$. In this scenario, the asymptotic behavior of $\{x_t\}_{t \geq 0}$ is captured by the ODE $\dot{x} = -\nabla_x F(x, y)\,\big|_{\,y = y^*(x)}$. Suppose $\min_x F(x, y)$ has a unique minimizer, denoted by $x^*(y)$, then $\{x_t, y_t\}_{t \geq 0}$ converges to some $(x^\natural, y^\natural)$ satisfying $x^\natural = x^*(y^\natural)$, $y^\natural = y^*(x^\natural)$. This implies that the two-timescale gradient update in (3.3) achieves the condition in (3.2) asymptotically. Moreover, when the minimizers of $F(\cdot, y)$ and $G(x, \cdot)$ are not unique, using similar analysis, it can be shown that $x^\natural$ is a local minimizer of $\min_x F(x, y^\natural)$, and simultaneously $y^\natural$ is a local minimizer of $\min_y G(x^\natural, y)$.

In the rest of this section, we apply the above argument to problems in reinforcement learning. In particular, we formulate both value function estimation and policy learning as bilevel optimization problems, which naturally motivates us to propose two-timescale algorithms with convergence guarantees.

### 3.1 VALUE FUNCTION ESTIMATION

In value function estimation, the goal is to estimate either $V^\pi$ or $Q^*$ defined in §2 using function approximation. Note that $V^\pi$ and $Q^*$ are the unique fixed points of Bellman operators $\mathcal{T}^\pi$ and $\mathcal{T}^*$, respectively. Here we mainly focus on $Q^*$, the results for $V^\pi$ can be similarly obtained by replacing $\mathcal{T}^*$ in (2.1) by $\mathcal{T}^\pi$.

Let $\mathcal{F} = \{Q_\theta \colon \mathcal{S} \times \mathcal{A} \to \mathbb{R}^d, \theta \in \mathbb{R}^d\}$ be a parametrized function class, where $\theta$ is the parameter. Our goal is to find $Q_\theta$ such that the mean-squared Bellman error (MSBE)

$$\ell(\theta) = \|Q_\theta - \mathcal{T}^* Q_\theta\|_\rho^2 = \mathbb{E}_{(s,a)\sim\rho}\Big(\big\{Q_\theta(s,a) - R(s,a) - \gamma \cdot \mathbb{E}\big[\max_{a'\in\mathcal{A}} Q_\theta(s',a') \,\big|\, s,a\big]\big\}^2\Big) \quad (3.5)$$

is minimized, where $\rho \in \mathcal{P}(\mathcal{S} \times \mathcal{A})$ is probability distribution, and $s' \sim P(\cdot \,|\, s,a)$ is the next state of the MDP. Note that $Q$-learning is an off-policy method. To learn $Q^*$, it is common to obtain sample from the MDP using a behavioral policy $\pi_b$, which induces a Markov chain on $\mathcal{S} \times \mathcal{A}$. Then $\rho$ can be viewed as the stationary distribution of this Markov chain. Furthermore, for simplicity, we assume that drawing i.i.d. samples from $\rho$ is possible, which approximately holds when the Markov chain enjoys rapid mixing properties.

Moreover, notice that in (3.5), the conditional expectation is inside the quadratic function, which is known to cause the issue of "double sampling" (Baird et al., 1995). That is, to obtain an unbiased estimate of the MSBE, given $(s,a) \sim \rho$, we need two conditionally independent samples from $\mathbb{P}(\cdot \,|\, s,a)$, which cannot be satisfied in practice. To avoid this issue, we write MSBE minimization as a bilevel optimization problem

$$\omega = \min_\omega \|\omega - \vartheta(\omega)\|_2^2 \qquad \text{subject to} \ \ \vartheta(\omega) = \operatorname*{argmin}_{\theta\in\mathbb{R}^d}\big\{L(\theta,\omega) = \|Q_\theta - \mathcal{T}^* Q_\omega\|_\rho^2\big\}. \quad (3.6)$$

We note that if there exists a $\theta^*$ such that $Q_{\theta^*} = Q^*$, then we have $\vartheta(\theta^*) = \theta^*$, which implies that $(\omega,\theta) = (\theta^*,\theta^*)$ is the solution of (3.6). Intuitively, with nonlinear function approximation, the lower level optimization problem $\min_\theta L(\theta,\omega)$ solves a nonlinear least square regression problem using function class $\mathcal{F}$, where $\mathcal{T}^* Q_\omega$ can be viewed as the response variable. After obtaining $\vartheta(\omega)$, then for the upper level optimization problem, we directly replace by $\vartheta(\omega)$. Thus, in the batch setting, solving the bilevel optimization in (3.6) yields the fitted $Q$-iteration algorithm (Riedmiller, 2005; Munos & Szepesvári, 2008). Specifically, When deep neural networks are used for function approximation, we recover the well-known deep $Q$-network (DQN) (Mnih et al., 2015), where $Q_\omega$ in (3.6) is the target network. We note that the usage of the target network is a critical trick in DQN, whose mechanism is not fully understood. Therefore, the target network in DQN can be viewed as the parameter of the upper level optimization of the bilevel formulation in (3.6). From the perspective of bilevel optimization, it is natural that the target network needs to be updated in a slower rate, which justifies the empirical practice of DQN that the target network is updated every $T_{\text{target}}$ iterations while the $Q$-network is updated in each iteration.

To obtain an online algorithm, we solve (3.6) via two-timescale gradient update given in (3.3). The update rules are given by $\omega_{t+1} \leftarrow (1-\alpha_t)\cdot\omega_t + \alpha_t\cdot\theta_t$, and $\theta_{t+1} \leftarrow \theta_t - \beta_t\cdot\nabla_\theta L(\theta_t,\omega_t)$. In addition, we note that $\nabla_\theta L(\theta,\omega)$ has an unbiased estimate

$$[Q_\theta(s,a) - R(s,a) - \gamma\cdot\max_{a'\in\mathcal{A}} Q_\omega(s',a')]\cdot\nabla_\theta Q_\theta(s,a).$$

Thus, replacing $\nabla_\theta L(\theta,\omega)$ by its estimate, we obtain our online $Q$-learning algorithm, which is stated in Algorithm 1. Here in line 6 we project the iterate to a compact set $\mathcal{X} = \{x \in \mathbb{R}^d \colon \|x\|_2 \le R_\mathcal{X}\}$ for stability, where $R_\mathcal{X}$ is a sufficiently large constant. It is clear that our algorithm is free from the double sampling issue. Moreover, as we see in §4, this algorithm converges almost surely to $\omega^*, \vartheta(\omega^*)$, where $\vartheta(\omega^*)$ is a local minimizer of $\min_\theta L(\theta,\omega^*)$, and $\omega^*$ is close to $\vartheta(\omega^*)$ up to some error caused by projection.

Finally, we note that Algorithm 1 can be similarly applied to policy evaluation, where we estimate $V^\pi$ using function class $\{V_\theta \colon \mathcal{S} \to \mathbb{R}, \omega \in \mathbb{R}^d\}$. In this case, by the definition of $\mathcal{T}^\pi$, we only need to replace $\delta_t$ in line 5 by $\delta_t = V_{\theta_t}(s_t) - R(s_t,a_t) - \gamma\cdot V_{\omega_t}(s_t')$. Moreover, here $\rho$ is set to be the stationary distribution induced by policy $\pi$. We defer the algorithm for policy evaluation to §A.

---

**Algorithm 1** Online Q-learning with nonlinear function approximation.
1: **Input:** Initialization $\theta_0 = \omega_0 \in \Theta$, stepsizes $\{\alpha_t, \beta_t\}_{t\geq 0}$.
2: **Output**: The sequences $\{\theta_t\}_{t\geq 0}$ and $\{\omega_t\}_{t\geq 0}$ .
3: **for** $t = 0, 1, 2, \dots$ **do**
4:     Sample $(s_t, a_t) \sim \rho$, observe reward $R(s_t, a_t)$ and next state $s_t'$.
5:     Compute the TD-error $\delta_t = Q_{\theta_t}(s_t, a_t) - R(s_t, a_t) - \gamma \cdot \max_{a\in\mathcal{A}} Q_{\omega_t}(s_t', a)$.
6:     Update the parameters by

$$\omega_{t+1} = \Pi_{\mathcal{X}}[(1 - \alpha_t) \cdot \omega_t + \alpha_t \cdot \theta_t], \qquad \theta_{t+1} = \theta_t - \beta_t \cdot \delta_t \cdot \nabla_\theta Q_{\theta_t}(s_t, a_t). \qquad (3.7)$$

7: **end for**

---

### 3.2 THE ACTOR-CRITIC ALGORITHM

Now we formulate policy learning as bilevel optimization and obtain the actor-critic algorithm. Let $\{\pi_\omega \colon \omega \in \mathbb{R}^p\}$ be the family of policies and let $\{V_\theta \colon \theta \in \mathbb{R}^d\}$ be a parametrized family of value functions. Note that the objective in policy learning can be written as $J(\omega) = \mathbb{E}_{s_0 \sim p_0}[V^{\pi_\omega}(s_0)]$. We define $F(\theta, \omega) = \mathbb{E}_{s_0 \sim p_0}[V_\theta(s_0)]$ and consider the bilevel optimization problem

$$\max_\omega F[\vartheta(\omega), \omega] \qquad \text{subject to} \quad \vartheta(\omega) = \operatorname*{argmin}_\theta \big\{ G(\theta, \omega) = \|V_\theta - \mathcal{T}^{\pi_\omega} V_\theta\|_{\rho^\omega}^2 \big\}, \qquad (3.8)$$

where $\rho^\omega$ is the state-occupancy measure of policy $\pi_\omega$. To see the correctness of such a formulation, notice that if $V^{\pi_\omega} \in \{V_\theta, \theta \in \mathbb{R}^d\}$, then the solution of the lower level optimization problem $\vartheta(\omega)$ satisfies that $V_{\vartheta(\omega)} = V^{\pi_\theta}$, which leads to $F[\vartheta(\omega), \omega] = J(\omega)$ for all $\omega \in \mathbb{R}^p$. Thus, we recover the original policy learning problem.

Note that $\min_\theta G(\theta, \omega)$ is an policy evaluation problem. Directly solving this problem suffers the issue of double sampling. To resolve this issue, as stated above, one need to formulate itself as a bilevel optimization problem. This will result in a three-timescale algorithm for policy learning. However, due to its complexity, instead we consider the TD(0) method, which is a semi-gradient method for policy evaluation (Sutton & Barto, 1998). Specifically, we define

$$g(\theta, \omega) = \mathbb{E}_{s\sim\rho^\omega, a\sim\pi_\omega(\cdot\,|\,s)} \big\{ [R(s, a) + \gamma \cdot V_\theta)(s') - V_\theta(s)] \cdot \nabla_\theta V_\theta(s) \big\}, \qquad (3.9)$$

which can be viewed as a biased estimate of $-\nabla_\theta G(\theta, \omega)$. Similar to (3.3), we approximately solve (3.8) via $\omega_{t+1} \leftarrow \omega_t + \alpha \nabla_\omega F(\theta_t, \omega_t)$, and $\theta_{t+1} \leftarrow \theta_t + \beta_t \cdot g(\theta_t, \omega_t)$. Moreover, by policy gradient theorem, we have

$$\nabla_\omega F(\theta, \omega) = \mathbb{E}_{s\sim\rho^\omega, a\sim\pi_\omega(\cdot\,|\,s)} \big\{ [R(s, a) + \gamma \cdot V_\theta)(s') - V_\theta(s)] \cdot \nabla_\omega \log \pi_\omega(a\,|\,s) \big\}. \qquad (3.10)$$

Thus, replacing (3.9) and (3.10) by their sample-based counterparts, we recover the actor-critic algorithm with TD(0) updates, whose details are deferred to §A. Moreover, for algorithmic stability, we project $\{\omega_t\}_{t\geq 0}$ to $\Omega = \{y \in \mathbb{R}^d \colon \|y\|_2 \leq R_\Omega\}$ with $R_\Omega$ sufficiently large.

Furthermore, we note that TD(0) with nonlinear function approximation can diverge in some situations (Tsitsiklis & Van Roy, 1997). In the next section, we given sufficient conditions for the convergence of actor-critic with nonlinear function approximation.

## 4 MAIN RESULTS

In this section, we establish the convergence results for the algorithm presented in §3. We first consider the online Q-learning algorithm with nonlinear function approximation. Theoretical results for policy evaluation can be similarly obtained, which are omitted for simplicity. We make the following assumption for the online Q-learning.

**Assumption 4.1.** For the online Q-learning algorithm, we assume the following conditions holds.

(i). For any $(s, a) \in \mathcal{S} \times \mathcal{A}$ and any $\theta \in \mathbb{R}^d$, we have $|Q_\theta(s, a)| \leq Q_{\max}$, $\|\nabla_\theta Q_\theta(s, a)\|_2 \leq G_{\max}$, and $\nabla_\theta Q_\theta(s, a)$ is Lipschitz continuous in $\theta$. Here $Q_{\max} \geq R_{\max}/(1 - \gamma)$ and $G_{\max}$ are two positive constants.

(ii). Let $L(\theta, \omega)$ be defined in (3.6). For any $\omega \in \mathcal{X}$, we assume that any local minimizer $\theta^*$ of $\min_\theta L(\theta, \omega)$ satisfies that $\nabla^2_\theta L(\theta^*, \omega) \succ 0$.

Here condition (i) in Assumption 4.1 requires that $Q_\theta(s, a)$, as a function of $\theta$, satisfies certain smoothness condition. Condition (ii) postulates that any local minimizer of $L_\omega(\theta) = L(\theta, \omega)$ can be determined using second order optimality condition. This assumption is strictly weaker than the strict saddle property (Ge et al., 2015), which is known to hold for a family of nonconvex functions.

**Theorem 4.2.** Under Assumptions 3.1 and 4.1, if $\sup_{t \geq 0} \|\theta_t\|_2 < \infty$, then $\{\theta_t, \omega_t\}_{t \geq 0}$ converges almost surely to some $(\theta^*, \omega^*)$, where $\omega^* \in \mathcal{X}$ and $\theta^*$ is a local minimizer of the function $L(\cdot, \omega^*)$. In addition, if $\|\omega^*\|_2 \leq R_\mathcal{X}$, we have $\theta^* = \omega^*$. Whereas if $\|\omega^*\|_2 = R_\mathcal{X}$, we have $\theta^* = \lambda \cdot \omega^*$ for some $\lambda \geq 1$.

To see the intuition of Theorem 4.2, consider the two-timescale gradient updates for the population problem given in (3.6). In the faster timescale, we could fix $\{\omega_t\}_{t \geq 0}$ at $\omega^*$. Then $\{\theta_t\}_{t \geq 0}$ converges to a local minimizer of $L(\cdot, \omega)$, which is denoted by $\vartheta(\omega^*)$. For the slower timescale, due to the projection $\Pi_\mathcal{X}$ in (3.7), the dynamics of $\{\omega_t\}_{t \geq 0}$ is characterized by a projected ODE $\dot{\omega} = \vartheta(\omega) - \omega + \xi^\omega$, where $\xi^\omega(t)$ is the additional term introduced by projection. Consider any asymptotically stable equilibrium $\omega^*$ of this projected ODE. If $\omega^*$ is in the interior of $\mathcal{X}$, then projection is not activated, and we have $\vartheta(\omega^*) = \omega^*$. If $\omega^*$ is on the boundary of $\mathcal{X}$, since $\mathcal{X}$ is a Euclidean ball, $\vartheta(\omega^*) - \omega^*$ must be in the same direction as $\omega^*$, which implies that $\vartheta(\omega^*) = \lambda \cdot \omega^*$ with $\lambda \geq 1$.

In the following, we lay out the assumption for the actor-critic algorithm.

**Assumption 4.3.** For the actor-critic algorithm, we assume that the following conditions holds.

(i). For any $s \in \mathcal{S}$ and any $\theta \in \mathbb{R}^d$, we assume that $|V_\theta(s)| \leq Q_{\max}$, $\|\nabla_\theta V_\theta(s)\|_2 \leq G_{\max}$, and $\nabla_\theta V_\theta(s)$ is Lipschitz continuous. In addition, there exists a constant $\pi_{\max} > 0$ such that $\|\nabla_\omega \log \pi_\omega(a \,|\, s)\|_2 \leq \pi_{\max}$ for all $(s, a) \in \mathcal{S} \times \mathcal{A}$ and $\omega \in \Omega$.

(ii). For function $g(\theta, \omega)$ defined in (3.9), we assume that for each $\omega \in \Omega$, the ODE $\dot{\theta}(t) = g[\theta(t), \omega]$ has a local asymptotically stable equilibrium $\vartheta(\omega)$. Moreover, $\vartheta(\cdot)$ is Lipschitz continuous in a neighborhood of $\omega$.

The first condition in Assumption 4.3 is parallel to that in Assumption 4.1. This condition ensures that both the value function and the policy function are regular. More importantly, as shown in §C.2, under the faster timescale, we could fix $\{\omega_t\}_{t \geq 0}$ at some $\omega^* \in \Omega$, and $\{\theta_t\}_{t \geq 0}$ converges to a stable equilibrium of ODE $\dot{\theta} = g(\theta, \omega^*)$. Thus, condition (ii) ensures that this equilibrium, as a function of $\omega$, is locally Lipschitz. We note that condition (ii) is more restrictive compared with the second condition in the previous assumption. The reason is that TD(0) update for the critic is not a gradient step, i.e., $g(\theta, \omega)$ is not equal to $\nabla_\theta G(\theta, \omega)$. Thus, $\{\theta_t\}_{t \geq 0}$ cannot converge to a local minimizer of $G(\cdot, \omega)$. Now we present the convergence result for the actor-critic algorithm.

**Theorem 4.4.** Under Assumptions 3.1 and 4.3, if $\sup_{t \geq 0} \|\theta_t\|_2 < \infty$, $\{\theta_t, \omega_t\}$ converges to some $\{\vartheta(\omega^*), \omega^*\}$, where $\vartheta(\omega^*)$ is an asymptotically stable equilibrium of ODE $\dot{\theta} = g(\theta, \omega^*)$. Moreover, if $\|\omega^*\|_2 < R_\Omega$, we have $\nabla_\omega F[\vartheta(\omega^*), \omega^*] = 0$. Whereas if $\|\omega^*\|_2 = R_\Omega$, then there exists $\lambda \geq 0$ such that $\nabla_\omega F[\vartheta(\omega^*), \omega^*] = \lambda \cdot \omega^*$.

Similar to Theorem 4.2, to determine the convergence of $\{\omega_t\}_{t \geq 0}$, we consider the slower timescale, under which we have the projected ODE $\dot{\omega} = \nabla_\omega F[\vartheta(\omega), \omega] + \xi^\omega$. By studying the equilibrium of this ODE, we obtain the above theorem. An interesting case is that when $\|\omega^*\|_2 < R_\Omega$, we have $g(\theta^*, \omega^*) = 0$ and $\nabla_\omega F(\theta^*, \omega^*) = 0$, where $\theta^* = \vartheta(\omega^*)$. Intuitively, this implies that actor-critic converges to a local Nash equilibrium.

## 5 NUMERICAL EXPERIMENTS

In this section, we evaluate the performance of our online Q-learning and actor-critic algorithms using OpenAI Gym (Brockman et al., 2016), which is a standard testbed for reinforcement learning.

**Online Q-learning.** First, we present the results for Algorithm 1 on Acrobot and MountainCar, which are two classical control tasks. The value functions for Acrobot and MountainCar are neural networks with one and two hidden layers, respectively, where all hidden layers have 64 neurons. To corroborate with our theory, in the experiments, we fix $\beta_t = 10^{-3}$ for all $t \geq 0$ and let $\alpha_t$ be various constants. We train the neural networks using minibatch stochastic gradient descent, where the batch size is 32. The results are plotted in Figure 1, where three independent experiments are performed for each setting. Here each curve is the mean value $\mu$ of the results in the same setting, together with the shadow representing the interval $[\mu - \sigma/2, \mu + \sigma/2]$, where $\sigma$ is the standard deviation. Also, in order to have a better visualization, we use a uniform moving average method with window size 20000 frames to smooth out the curves.

As shown in Figure 1, for both Acrobot and MountainCar, when $\alpha_t < \beta_t$ (green and blue lines), the reward grows gradually and approaches the maximum reward. Whereas if $\alpha_t > \beta_t$ (dark yellow and black lines), the reward finally approaches the minimum. This sharp transition justifies the validity of using two-timescale updates with $\beta_t \gg \alpha_t$ in the online Q-learning.

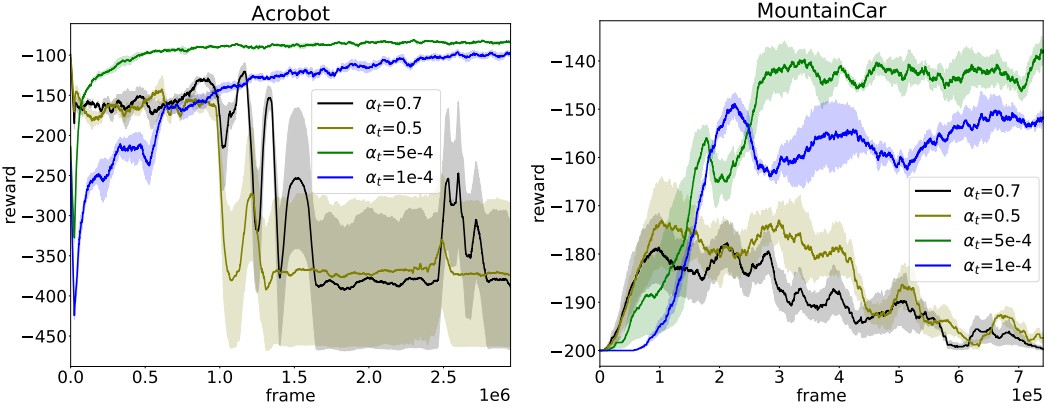

Figure 1: Training Acrobot and MountainCar using online Q-learning. Here we set $\beta_t = 10^{-3}$ for both problems. In addition, for Acrobot, we let $\alpha_t$ be one of $\{10^{-2}, 8 \cdot 10^{-3}, 3 \cdot 10^{-4}, 10^{-4}\}$. For or MountainCar we set $\alpha_t$ to be one of $\{0.7, 0.5, 5 \cdot 10^{-4}, 10^{-4}\}$.

**Two-timescale actor-critic algorithm.** We also test the actor-critic algorithm on two Atari 2600 games: Pong and Breakout. Here we implement our algorithm based on the A3C framework (Mnih et al., 2016) with 32 asynchronous agents in parallel. In these experiments, we fix stepsize of the actor to $\alpha_t = 10^{-4}$ and let the stepsize of the actor $\beta_t$ be various constants. Similar to the previous experiment, we plot the results in Figure 2 based on three independent trials, which is deferred to the appendix.

As shown in Figure 2, for those settings with a larger critic stepsize $\beta_t$ (green and blue lines), the reward gradually converges to its maximum, while for others with a smaller critic stepsize (dark yellow and black lines), the rewards stay at a very low level. This agrees with our claim in Theorem 4.4 that the critic proceeds in a faster speed than the actor.

## 6 CONCLUSION

We study online reinforcement learning with nonlinear function approximation in general. Using the unified framework of bilevel optimization, we propose online first-order algorithms for both value function estimation and policy learning. Moreover, using stochastic approximation results, we show that the asymptotic behavior of the algorithms are captured by ordinary differential equations. Finally, we perform thorough empirical studies in support of our theory.

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

## A   THE POLICY EVALUATION AND ACTOR-CRITIC ALGORITHMS

---

**Algorithm 2** Online policy evaluation with nonlinear function approximation.

---

1: **Input:** Initialization $\theta_0 = \omega_0 \in \Theta$, stepsizes $\{\alpha_k\}$, $\{\beta_k\}$, and sampling distribution $\rho \in \Delta(\mathcal{S} \times \mathcal{A})$.
2: **Output**: The sequences $\{\theta_t\}_{t \geq 0}$ and $\{\omega_t\}_{t \geq 0}$ .
3: **for** $t = 0, 1, 2, \ldots$ **do**
4:    Sample $(s_t, a_t) \sim \rho$, observe reward $R(s_t, a_t)$ and next state $s'_t$.
5:    Compute the TD error $\delta_t = Q_{\theta_t} - R_t - \gamma \cdot \max_{a \in \mathcal{A}} Q_{\omega_t}(S'_t, a)$.
6:    Update the function estimates by

$$\theta_{t+1} = \Pi_\Omega[(1 - \alpha_t) \cdot \omega_t + \alpha_t \cdot \theta_t], \qquad \theta_{t+1} = \theta_t - \beta_t \cdot \delta_t \cdot \nabla_\theta Q_{\theta_t}(S_t, A_t). \quad (A.1)$$

7: **end for**

---

---

**Algorithm 3** The actor-critic algorithm with nonlinear function approximation.

---

1: **Input:** Initialization $\omega_0 \in \mathbb{R}^d$ and $\theta_0 \in \mathbb{R}^p$, stepsizes $\{\alpha_t, \beta_t\}_{t \geq 0}$.
2: **Output**: The sequences $\{\theta_t\}_{t \geq 0}$ and $\{\omega_t\}_{t \geq 0}$ .
3: **for** $t = 0, 1, 2, \ldots$ **do**
4:    Sample $s_t \sim \rho^{\omega_t}$, take action $a_t \sim \pi_{\omega_t}(\cdot \mid s_t)$, observe reward $R(s_t, a_t)$ and the next state $s'_t$.

5:    Compute the TD-error $\delta_t = R(s_t, a_t) + \gamma \cdot V_{\theta_t}(s'_t) - V_{\theta_t}(s_t)$.
6:    Update the parameters by

$$\omega_{t+1} = \Pi_\Omega[\omega_t + \alpha_t \cdot \nabla_\omega \log \pi_{\omega_t}(a_t \mid s_t) \cdot \delta_t], \qquad \theta_{t+1} = \theta_t + \beta_t \cdot \delta_t \cdot \nabla_\theta V_{\theta_t}(s_t). \quad (A.2)$$

7: **end for**

---

## B   ADDITIONAL FIGURES OF NUMERICAL EXPERIMETNS

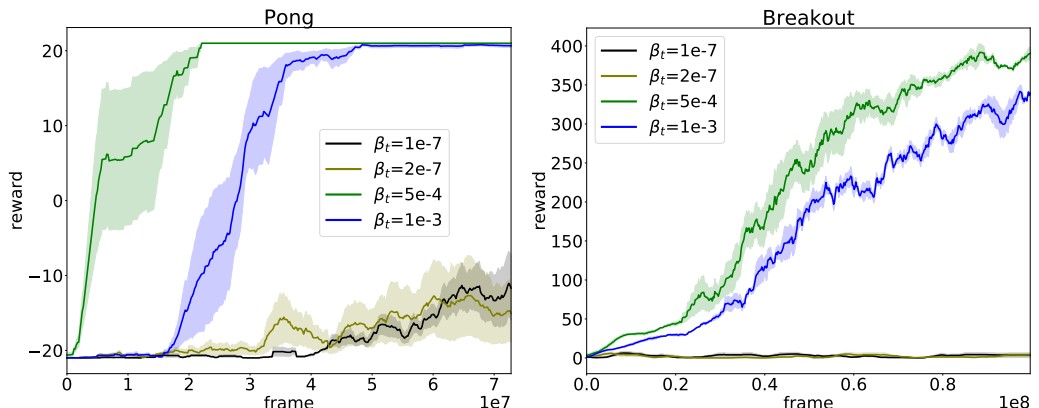

Figure 2: Training Pong and Breakout using two timescale actor-critic algorithm, where we use 32 agents, minibatch of size 20, with a fixed actor network stepsize $\alpha_t \equiv 10^{-4}$, but a varied critic network stepsize $\beta_t \in \{10^{-7}, 2 \cdot 10^{-7}, 5 \cdot 10^{-4}, 10^{-3}\}$. Left: training Pong. Right: training Breakout.

## C   PROOFS OF THE MAIN RESULTS

In this section, we lay out the proofs of Theorems 4.2 and 4.4.

C.1 PROOF OF THEOREM 4.2

*Proof.* Our proof consists of two steps. The first step consists of the analysis under the faster timescale, where we show that in this case we could fix $\{\omega_t\}_{t\geq 0}$ at some $\omega^* \in \mathcal{X}$ and only focus on $\{\theta_t\}_{t\geq 0}$. In the second step, we look into the slower timescale, which yields the convergence result for $\{\omega_t\}_{t\geq 0}$ and completes the proof.

**Step 1. Faster timescale.** We first consider the convergence of $\{\theta_t, \omega_t\}_{t\geq 1}$ in the faster timescale. Using stochastic approximation, we will show that the sequence $\{\theta_t\}_{t\geq 0}$ defined in (3.7) tracks a local minimizer of the lower level optimization.

To begin with, for notational simplicity, we define

$$\delta_t = Q_{\theta_t}(s_t, a_t) - r(s_t, a_t) - \gamma \cdot Q_{\omega_t}(s_t', a_t'), \quad \phi_t = \nabla_\theta Q_{\theta_t}(s_t, a_t),$$

for all $t \geq 0$, where $a_t' = \mathrm{argmax}_{a \in \mathcal{A}} Q_{\theta_t}(s_t', a)$. Then the updating rules in Algorithms 1 reduce to

$$\omega_{t+1} \leftarrow \Pi_{\mathcal{X}}\big[\omega_t + \alpha_t \cdot (\theta_t - \omega_t)\big], \qquad \theta_{t+1} \leftarrow \theta_t - \beta_t \cdot \delta_t \cdot \phi_t, \tag{C.1}$$

where $\{\alpha_t, \beta_t\}_{t\geq 0}$ are the stepsizes, and $\Pi_{\mathcal{X}}$ is the projection operator onto set $\mathcal{X}$, which is assumed to be a closed Euclidean ball with radius $R_\omega$. In addition, let $\mathcal{F}_t^{(1)} = \sigma(\{\theta_\ell, \omega_\ell\}_{\ell \leq t})$ be the $\sigma$-algebra generated by the parameter iterates until time $t$. By definition, $\theta_t$ and $\omega_t$ are $\mathcal{F}_t^{(1)}$-measurable for any $t \geq 1$. Furthermore, for any $\theta \in \mathbb{R}^d$ and $\omega \in \mathbb{R}^p$, we define

$$h(\theta, \omega) = \mathbb{E}_{(s,a)\sim\rho}\big\{[Q_\theta(s,a) - (\mathcal{T}^*Q_\omega)(s,a)] \cdot \nabla_\theta Q_\theta(s,a)\big\}, \tag{C.2}$$

Thus, by the definition of the Bellman optimality operator in , for any $t \geq 0$, we have $\mathbb{E}[\delta_t \phi_t \,|\, \mathcal{F}_t^{(1)}] = h(\theta_t, \omega_t)$. This implies that $\{\zeta_t\}_{t\geq 0}$ defined by $\zeta_t = h(\theta_t, \omega_t) - \delta_t \cdot \phi_t$ is a martingale difference sequence with respect to the filtration $\{\mathcal{F}_t^{(1)}\}_{t\geq 0}$.

Moreover, for any compact set $\mathcal{X} \subseteq \mathbb{R}^d$, we denote by $\mathcal{C}_{\mathcal{X}}(x)$ the outer normal cone of $\mathcal{X}$ at $x \in \mathcal{X}$. That is, $\mathcal{C}_{\mathcal{X}}(x) = \{u \in \mathbb{R}^d \colon u^\top(y - x) \leq 0, \forall y \in \mathcal{X}\}$. In particular, if $x$ is in the interior of $\mathcal{X}$, then $\mathcal{C}_{\mathcal{X}}(x)$ is empty. Moreover, since we assume $\mathcal{X}$ is an Euclidean ball with radius $R_{\mathcal{X}}$, for any $x$ in the boundary of $\mathcal{X}$, denoted by $\partial\mathcal{X}$, we have $\mathcal{C}_{\mathcal{X}}(x) = \{\lambda \cdot x \colon \lambda \geq 0\}$. Using the notion of the outer normal cone, the first equation in (C.1) can be written as

$$\omega_{t+1} = \omega_t + \alpha_t \cdot (\theta_t - \omega_t + \xi_t), \tag{C.3}$$

where $\xi_t \in -\mathcal{C}_{\mathcal{X}}(\omega_{t+1})$ is the correction term caused by projection onto $\mathcal{X}$. Since $(1 - \alpha_t) \cdot \omega_t \in \mathcal{X}$ and $\omega_{t+1}$ is the projection of $\omega_t + \alpha_t \cdot (\theta_t - \omega_t)$, we have

$$\|\xi_t\|_2 = 1/\alpha_t \cdot \inf_{y\in\mathcal{X}} \|\omega_t + \alpha_t \cdot (\theta_t - \omega_t) - y\|_2 \leq \|\theta_t\|_2 \tag{C.4}$$

for all $t \geq 0$. Then, writing the two equations in (C.1) together, we have

$$\begin{pmatrix} \theta_{t+1} \\ \omega_{t+1} \end{pmatrix} = \begin{pmatrix} \theta_t \\ \omega_t \end{pmatrix} + \beta_t \cdot \begin{pmatrix} -h(\theta_t, \omega_t) + \zeta_t \\ \alpha_t/\beta_t \cdot (\theta_t - \omega_t + \xi_t) \end{pmatrix}. \tag{C.5}$$

In the following, we apply the result on ODE approximation to (C.5). First note that under Assumption 4.1, $|Q_\theta(s,a)|$, $\|\nabla_\theta Q_\theta(s,a)\|_2$ are bounded by $Q_{\max}$ and $G_{\max}$, respectively. This implies that implies that $\|\delta_t \cdot \phi_t\|$ is bounded by $2Q_{\max} \cdot G_{\max}$ for all $t \geq 1$. Thus, $\{\zeta_t\}_{t\geq 0}$ is a bounded martingale sequence. Then there exist an absolute constant $C > 0$ such that

$$\mathbb{E}[\|\zeta_t\|_2^2 \,|\, \mathcal{F}_t] \leq C \qquad \text{for all} \quad t \geq 1.$$

Moreover, let $M_T = \sum_{t=0}^{T} \beta_t \cdot \zeta_t$ for any $T \geq 0$. Since $\sum_{t\geq 0} \beta_t^2 < \infty$, by the Martingale convergence theorem (Proposition VII-2-3(c) on page 149 of Neveu (1975)), $\{M_T\}_{T\geq 0}$ converges almost surely. Additionally, $M_T$ is square integrable with $\mathbb{E}\|M_T\|_2^2 \leq C \cdot \sum_{t\geq 0} \beta_t^2$. Thus, for any $\epsilon > 0$, by Doob's martingale inequality (Doob, 1953), we have

$$\mathbb{P}\Big( \sup_{N\geq T} \|M_N - M_T\| \geq \epsilon \Big) \leq \frac{\sup_{N\geq T} \mathbb{E}[\|M_N - M_T\|_2^2]}{\epsilon^2} \leq \frac{2C^2 \sum_{t\geq T} \beta_t^2}{\epsilon^2},$$

which converges to zero as $T$ goes to infinity. Furthermore, under Assumption 3.1 and the assumption that $\sup_{t\geq 0}\|\theta_t\|_2 < \infty$, it holds that $\alpha_t/\beta_t \cdot (\theta_t - \omega_t + \xi_t) = 0$ as $t$ goes to infinity. Besides, since both $\nabla_\theta Q_\theta(s,a)$ and $Q_\theta(s,a)$ are bounded and Lipschitz continuous under Assumption 4.1, $h(\theta,\omega)$ defined in (C.2) is Lipschitz in both $\theta$ and $\omega$.

Now we apply Theorem D.2 to sequence $\{(\theta_t,\omega_t)\}_{t\geq 0}$, which implies that the asymptotic behavior of $\{(\theta_t,\omega_t)\}_{t\geq 0}$ converges almost surely to the set of asymptotically stable equilibria of the ODE

$$\dot\theta = -h(\theta,\omega), \qquad \dot\omega = 0. \tag{C.6}$$

Specifically, the set of asymptotically stable equilibria of the ODE in (C.6), denoted by $\mathcal{K}_1$, is

$$\mathcal{K}_1 = \big\{(\theta^*,\omega^*)\colon \omega^* \in \mathcal{X}, h(\theta^*,\omega^*) = 0\big\}. \tag{C.7}$$

For any $\omega \in \mathbb{R}^d$, we denote $L_\omega(\theta) = L(\theta,\omega) = \|Q_\theta - \mathcal{T}^*Q_\omega\|_\rho^2$. Since $h(\theta,\omega) = \nabla_\theta L_{\omega^*}(\theta)$, the asymptotically stable equilibrium of ODE $\dot\theta = -\nabla_\theta L_{\omega^*}(\theta)$ is the local minima of $L_{\omega^*}(\theta)$.

Therefore, for the analysis under the faster timescale, essentially we can fix $\{\omega_t\}_{t\geq 0}$ at some $\omega^*$ and consider solely the asymptotic behavior of $\{\theta_t\}_{t\geq 0}$. In this case, $\{\theta_t\}_{t\geq 0}$ converges almost surely to a local minimizer $\theta^*$ of $L_{\omega^*}(\cdot)$. We note that $L_{\omega^*}(\cdot)$ may have multiple local minimizers. In this case, the local minimizer that $\{\theta_t\}_{t\geq 0}$ converges to is determined by the basin of attraction that $\{\theta_t\}_{t\geq 0}$ enters. Moreover, note that under Assumption 4.1, $\nabla_\theta^2 L_\omega(\cdot)$ is positive definite at all its local minima, which implies that the set of all local minima of $L_{\omega^*}(\cdot)$ is a disjoint set.

Consider the equation $\nabla_\theta L_{\omega^*}(\theta) = 0$. Note that $\theta = \theta^*$ is a solution. Besides, by the continuity of $\nabla_\theta^2 L_\omega(\cdot)$, there exists a open neighborhood $\mathcal{U}_\theta$ of $\theta^*$ and a positive number $C_L$ such that

$$\nabla_\theta^2 L_{\omega^*}(\theta) \succ C_L \cdot I_d \qquad \forall \theta \in \mathcal{U}_\theta.$$

By Lipschitz implicit function theorem (Dontchev & Hager, 1994), there exist a neighborhood $\mathcal{U}$ of $\theta^*$, a neighborhood $\mathcal{V}$ of $\omega^*$, and a mapping $\vartheta\colon \mathcal{V} \to \mathcal{U}$ such that $\vartheta(\omega^*) = \theta^*$. In addition, $\vartheta$ is Lipschitz continuous and satisfies $\nabla_\theta L(\theta,\omega)|_{\theta=\vartheta(\omega)} = 0$ for all $\omega \in \mathcal{V}$.

Thus, we show that, in the faster timescale, $\{\theta_t,\omega_t\}_{t\geq 0}$ converges almost surely to $[\vartheta(\omega^*),\omega^*]$, where $\omega^* \in \mathcal{X}$ and $\vartheta$ is Lipschitz continuous.

**Step 2. Slower timescale.**

Note that (C.7) cannot characterize the convergence of $\{\omega_t\}_{t\geq 0}$. Now we look into the dynamics in the slower timescale for a finer characterization. Recall that the update rule of the $\{\omega_t\}_{t\geq 0}$ is given in (C.3). For ease of presentation, we define $\sigma$-field $\mathcal{F}_t^{(2)} = \sigma(\{\omega_\ell\}_{\ell\leq t})$ for all $t \geq 0$. b In addition, we define

$$\psi_t = \mathbb{E}\big[(\theta_t - \omega_t)\,|\,\mathcal{F}_t^{(2)}\big] - [(\vartheta(\omega_t) - \omega_t] = \mathbb{E}\big[\theta_t\,|\,\mathcal{F}_t^{(2)}\big] - \vartheta(\omega_t). \tag{C.8}$$

Then (C.3) becomes $\omega_{t+1} = \omega_t + \alpha_t \cdot [\vartheta(\omega_t) - \omega_t] + \alpha_t \cdot (\psi_t + \xi_t)$, where $\xi_t \in -\mathcal{C}_\mathcal{X}(\omega_{t+1})$.

As we have shown in the first step of the proof, $\vartheta(\omega_t) - \theta_t$ converges to zero as $t$ goes to infinity, where the mapping $\vartheta\colon \mathbb{R}^d \to \mathbb{R}^d$ is Lipschitz continuous. In addition, since $\sup_{t\geq 0}\|\theta_t\|_2$ is finite, $\psi_t$ defined in (C.8) is uniformly bounded for all $t \geq 0$.

Now we apply the Kushner-Clark lemma (Kushner & Clark, 1978, see also Theorem E.2 in §E for details.) to sequence $\{\omega_t\}_{t\geq 0}$, it holds that $\{\theta_t\}_{t\geq 1}$ converges almost surely to the set of asymptotically stable equilibria of the ODE

$$\dot\omega = \vartheta(\omega) - \omega + \xi^\omega, \qquad \xi^\omega(t) \in -\mathcal{C}_\mathcal{X}(\omega(t)), \tag{C.9}$$

where the function $\xi^\omega$ appears due to the projection in (C.11). Recall that $\mathcal{X}$ is a Euclidean ball with radius $R_\mathcal{X}$. For any asymptotically stable equilibrium $\omega^*$ of (C.9), if $\omega^*$ is in the interior of $\mathcal{X}$, we have $\mathcal{C}_\mathcal{X}(\omega^*) = 0$, which implies that $\omega^* = \vartheta(\omega^*)$. In this case, sequence $\{(\theta_t,\omega_t)\}_{t\geq 1}$ converges almost surely to $(\omega^*,\omega^*)$, which satisfies that

$$\mathbb{E}_{(s,a)\sim\rho}\big\{[Q_{\omega^*}(s,a) - (\mathcal{T}^*Q_{\omega^*})(s,a)] \cdot \nabla_\theta Q_{\omega^*}(s,a)\big\} = 0.$$

Meanwhile, if $\omega^* \in \mathcal{X}$, we have $\vartheta(\omega^*) - \omega^* \in \mathcal{C}_{\mathcal{X}}(\omega^*)$, which implies that there exists $\lambda > 0$ such that $\vartheta(\omega^*) = \lambda \cdot \theta^*$. Thus, $\{(\theta_t, \omega_t)\}_{t \geq 1}$ converges almost surely to $(\lambda \cdot \omega^*, \omega^*)$. Therefore, we conclude the proof of Theorem 4.2. $\qquad\square$

## C.2 PROOF OF THEOREM 4.4

*Proof.* The proof is similar to that of Theorem 4.2 in §C.1. We prove the theorem in two steps, considering the faster and slower timescales seperately.

**Step 1. Faster timescale.** We first consider the convergence of $\{\theta_t, \omega_t\}_{t \geq 1}$ in the faster timescale. Using stochastic approximation, we will show that the critic sequence $\{\theta_t\}_{t \geq 0}$ converges to an asymptotically stable equilibrium of ODE $\dot{\theta} = g(\theta.\omega^*)$ for some $\omega^* \in \Omega$.

First recall that, for any $t \geq 0$, we have $s_t \sim \rho^{\omega_t}$, $a_t \sim \pi_{\omega_t}(\cdot \mid s_t)$, and $s_t' \sim P(\cdot \mid s_t, a_t)$. To simplify the notation, we define

$$\delta_t = V_{\theta_t}(s_t) - r(s_t, a_t) - \gamma \cdot V_{\theta_t}(s_t'), \quad \phi_t = \nabla_\theta V_{\theta_t}(s_t), \quad A_t = \nabla_\omega \log \pi_{\omega_t}(a_t \mid s_t). \quad \text{(C.10)}$$

Using these terms, the updating rules in Algorithm 3 become

$$\omega_{t+1} \leftarrow \Pi_\Omega[\omega_t + \alpha_t \cdot \delta_t \cdot A_t], \qquad \theta_{t+1} \leftarrow \theta_t + \beta_t \cdot \delta_t \cdot \phi_t, \qquad \text{(C.11)}$$

where $\{\alpha_t, \beta_t\}_{t \geq 0}$ are the stepsizes, and $\Pi_\Omega$ is the projection operator onto $\Omega$. In addition, under Assumption 4.3, we have $|V_\theta(s)| \leq V_{\max}$ and $\|\nabla_\omega \log \pi_\omega(a \mid s)\|_2 \leq \pi_{\max}$. Thus $\delta_t$ and $A_t$ defined in (C.10) satisfy $|\delta_t| \leq 2V_{\max}$ and $\|A_t\| \leq \pi_{\max}$ for all $t \geq 1$. Combining (C.11) and Assumption 3.1, this implies that $\alpha_t/\beta_t \cdot \delta_t \cdot A_t$ converges to zero as $t$ goes to infinity.

Moreover, let $\mathcal{F}_t^{(1)} = \sigma(\{\theta_\ell, \omega_\ell\}_{\ell \leq t})$. Recall that we define function $g(\theta, \omega)$ in (3.9). To simplify the notation, let $h(\theta, \omega) = \nabla_\omega F(\theta, \omega$, which is defined in (3.10). Then by (C.10), we have

$$\mathbb{E}[\delta_t \cdot \phi_t \mid \mathcal{F}_t^{(1)}] = g(\theta_t, \omega_t), \qquad \mathbb{E}[\delta_t \cdot A_t \mid \mathcal{F}_t^{(1)}] = h(\theta_t, \omega_t), \qquad \text{(C.12)}$$

which implies that $\{\zeta_t\}_{t \geq 0}$ is a martingale difference sequence, where we define $\zeta_t = \delta_t \cdot \phi_t - g(\theta_t, \omega_t)$ for all $t \geq 0$. Moreover, under Assumption 4.3, we have $\|\delta_t \cdot \phi_t\|_2 \leq 2V_{\max} \cdot G_{\max}$, which implies that there exists an absolute constant $C$ such that $\mathbb{E}[\|\zeta_t\|_2^2] < C$. Therefore, similar to the proof of Theorem 4.2 in §C.1, by applying Theorem D.2, we obtain that, in the faster timescale, sequence $\{\theta_t, \omega_t\}_{t \geq 0}$ converges almost surely to the asymptotically stable equilibria of ODE system $\{\dot{\theta} = g(\theta, \omega), \dot{\omega} = 0\}$. Under Assumption 4.3, this implies that $\{\omega_t\}_{t \geq 0}$ converges to some $\omega^*$, while $\{\theta_t\}_{t \geq 0}$ converges to a local asymptotically stable equilibrium $\theta^* = \vartheta(\omega^*)$. Moreover, $\vartheta \colon \mathbb{R}^p \to \mathbb{R}^p$ is Lipschitz continuous in a open neighborhood of $\omega^*$.

**Step 2. Slower timescale.** In the sequel, we characterize the convergence of $\{\theta_t, \omega_t\}_{t \geq 1}$ under the slower timescale. To begin with, we define $\mathcal{F}_t^{(2)} = \sigma(\{\omega_\ell\}_{\ell \leq t})$ for all $t \geq 0$. In addition, we define

$$\psi_t^{(1)} = -\delta_t \cdot A_t + \mathbb{E}[\delta_t \cdot A_t \mid \mathcal{F}_t^{(2)}], \qquad \psi_t^{(2)} = -\mathbb{E}[\delta_t \cdot A_t \mid \mathcal{F}_t^{(2)}] + h[\theta_t, \omega(\theta_t)], \qquad \text{(C.13)}$$

where function $g$ is defined in (3.9), $A_t$ and $\phi_t$ are defined in (C.10). Since $\|\delta_t \cdot A_t\|_2 \leq 2V_{\max} \cdot G_{\max}$, $\{\psi_t^{(1)}\}_{t \geq 1}$ is a bounded martingale difference sequence. Moreover, since $\mathcal{F}_t^{(2)} \subseteq \mathcal{F}_t^{(1)}$, by the tower property and (C.12), we have

$$\mathbb{E}[\delta_t \cdot A_t \mid \mathcal{F}_t^{(2)}] = \mathbb{E}[\mathbb{E}(\delta_t \cdot A_t \mid \mathcal{F}_t^{(1)}) \mid \mathcal{F}_t^{(2)}] = \mathbb{E}[h(\theta_t, \omega_t) \mid \mathcal{F}_t^{(2)}].$$

Using the notation in (C.13), the primal update in (C.11) can be written as

$$\omega_{t+1} = \Pi_\Omega[\omega_t + \alpha_t \cdot h[\omega_t, \vartheta(\omega_t)] + \alpha_t \cdot \psi_t^{(1)} + \alpha_t \cdot \psi_t^{(2)}], \qquad \text{(C.14)}$$

where $\vartheta(\omega^*)$ is a local asymptotically stable attractor of ODE $\dot{\theta} = h(\theta, \omega^*)$. Moreover, under Assumption 4.3, $\nabla_\theta V_\theta(s)$ and $\nabla_\omega \log \pi_\omega(a \mid s)$ are bounded, which implies that

$$\|h(\theta^{(1)}, \omega) - h(\theta^{(2)}, \omega)\| \leq 2G_{\max} \cdot \pi_{\max}$$

for any $\theta^{(1)}, \theta^{(2)} \in \mathbb{R}^d$ and any $\omega \in \Omega$. Then, by Cauchy-Schwarz inequality, we have

$$\|\psi_t^{(2)}\|_2 \leq \mathbb{E}\big\{\big\|h(\theta_t, \omega_t) - h[\vartheta(\omega_t), \omega_t]\big\|_2 \,\big|\, \mathcal{F}_t^{(2)}\big\} \leq 2G_{\max} \cdot \pi_{\max} \cdot \mathbb{E}\big[\|\theta_t - \vartheta(\omega_t)\|_2 \,\big|\, \mathcal{F}_t^{(2)}\big],$$

which tends to zero as $t$ goes to infinity. As shown in the first step of the proof, $\vartheta(\omega_t) - \omega_t$ converges to zero as $t$ goes to infinity.

Furthermore, we define $W_T = \sum_{t=1}^T \alpha_t \cdot \psi_t^{(1)} \in \mathbb{R}^d$ for any $T \geq 1$. Since $\sum_{t \geq 1} \alpha_t^2 < \infty$ and $\psi_t^{(1)}$ is bounded, $\{W_T\}_{T \geq 1}$ is a square-integrable martingale sequence, which converges almost surely by the Martingale convergence theorem (Neveu, 1975). Moreover, Doob's martingale inequality (Doob, 1953) implies that

$$\lim_{T \to \infty} \mathbb{P}\Big(\sup_{N \geq T} \|W_N - W_T\| \geq \epsilon\Big) \leq \lim_{T \to \infty} \frac{\sup_{N \geq T} \mathbb{E}[\|W_N - W_T\|_2^2]}{\epsilon^2} \leq \lim_{T \to \infty} \frac{2C^2 \sum_{t \geq T} \alpha_t^2}{\epsilon^2} = 0.$$

Finally, applying by the Kushner-Clark lemma (Kushner & Clark, 1978) to sequence $\{\omega_t\}_{t \geq 0}$, it holds that $\{\omega_t\}_{t \geq 0}$ converges almost surely to the set of asymptotically stable equilibria of the ODE

$$\dot{\omega} = h[\vartheta(\omega), \omega] + \xi^\omega, \qquad \xi^\omega(t) \in -\mathcal{C}_\Omega(\omega(t)), \tag{C.15}$$

where $\mathcal{C}_\Omega(\omega)$ is the outer normal cone of $\Omega$ at $\omega$. Recall that we assume that $\Omega$ is a Euclidean ball with radius $R_\Omega$. Thus, for any asymptotically stable equilibrium $\omega^*$ of (C.9), if $\omega^*$ is in the interior of $\Omega$, i.e., $\|\omega^*\| < R_\Omega$, we have $h[\vartheta(\omega^*), \omega^*] = 0$. Additionally, if $\|\omega^*\|_2 = R_\Omega$, then there exists $\lambda \geq 0$ such that $h[\vartheta(\omega^*), \omega^*] = \lambda \cdot \omega^*$. Therefore, we conclude the proof of Theorem 4.4. $\qquad\square$

## D   BACKGROUND ON STOCHASTIC APPROXIMATION

We first present a fundamental result for stochastic approximation which is obtained from Borkar (2008). Consider a sequence of iterations in $\mathbb{R}^d$

$$x_{t+1} = x_t + \gamma_t \cdot [G(x_t) + M_{t+1}], \quad t \geq 0, \quad x_0 \in \mathbb{R}^d. \tag{D.1}$$

Here $G \colon \mathbb{R}^d \to \mathbb{R}^d$ is a deterministic mapping, $M_{t+1} \in \mathbb{R}^d$ is a random vector, and $\gamma_t > 0$ is the stepsize.

**Assumption D.1.** We make the following assumption on the iteration (D.1).

- Function $G \colon \mathbb{R}^d \to \mathbb{R}^d$ is Lipschitz continuous.

- The stepsizes $\{\gamma_t\}_{t \geq 0}$ satisfies $\sum_{t \geq 0} \gamma_t = \infty$ and $\sum_{t \geq 0} \gamma_t^2 < \infty$;

- Random vectors $\{M_t\}_{t \geq 0}$ is a martingale difference sequence. That is, $M_0 = 0$, $\mathbb{E}[M_{t+1} \,|\, x_\tau, M_\tau, \tau \leq t] = 0$. Moreover, we assume that there exists some $K > 0$ such that

$$\mathbb{E}\big(\|M_{t+1}\|^2 \,|\, x_\tau, M_\tau, \tau \leq t\big) \leq K \cdot (1 + \|x_t\|^2).$$

Then the asymptotic behavior of the $\{x_t\}_{t \geq 0}$ in (D.1) is characterized by ODE

$$\dot{x} = G(x) \tag{D.2}$$

Suppose Eq. (D.2) has a unique globally asymptotically stable equilibrium $x^*$, we then have the following two theorems.

**Theorem D.2.** Under Assumption D.1, if $\sup_t \|x_t\|_2 < \infty$ almost surely, we have $x_t \to x^*$ almost surely.

**Theorem D.3.** Under Assumption D.1, suppose

$$\lim_{c \to \infty} \frac{G(cx)}{c} = G_\infty(x)$$

exists uniformly on compact sets for some $G_\infty \in C(\mathbb{R}^n)$. If the ODE $\dot{y} = G_\infty(y)$ has origin as the unique globally asymptotically stable equilibrium, then we have

$$\sup_t \|x_t\| < \infty \quad \text{almost surely}.$$

# E  KUSHNER-CLARK LEMMA

We state here the well-known Kushner-Clark Lemma (Kushner & Clark, 1978; Metivier & Priouret, 1984; Prasad et al., 2014) in the sequel.

Let $\Gamma$ be an operator that projects a vector onto a compact set $\mathcal{X} \subseteq \mathbb{R}^N$. Define a vector $\widehat{\Gamma}(\cdot)$ as

$$\widehat{\Gamma}[h(x)] = \lim_{0 < \eta \to 0} \left\{ \frac{\Gamma[x + \eta h(x)] - x}{\eta} \right\},$$

for any $x \in \mathcal{X}$ and with $h : \mathcal{X} \to \mathbb{R}^N$ continuous. Consider the following recursion in $N$ dimensions

$$x_{t+1} = \Gamma \big\{ x_t + \gamma_t [h(x_t) + \xi_t + \beta_t] \big\}. \tag{E.1}$$

The ODE associated with (E.1) is given by

$$\dot{x} = \widehat{\Gamma}[h(x)]. \tag{E.2}$$

**Assumption E.1.**   We make the following assumptions:

- $h(\cdot)$ is a continuous $\mathbb{R}^N$-valued function.

- The sequence $\{\beta_t\}$, $t \geq 0$ is a bounded random sequence with $\beta_t \to 0$ almost surely as $t \to \infty$.

- The stepsizes $\gamma_t$, $t \geq 0$ satisfy $\gamma_t \to 0$ as $t \to \infty$ and $\sum_t \gamma_t = \infty$.

- The sequence $\xi_t$, $t \geq 0$ satisfies for any $\epsilon > 0$

$$\lim_t \mathbb{P}\left( \sup_{n \geq t} \left\| \sum_{\tau=t}^n \gamma_\tau \xi_\tau \right\|_2 \geq \epsilon \right) = 0.$$

Then the Kushner-Clark Lemma says the following.

**Theorem E.2.**  Under Assumption E.1, suppose that the ODE (E.2) has a compact set $\mathcal{K}^*$ as its set of asymptotically stable equilibria. Then $x_t$ in (E.1) converges almost surely to $\mathcal{K}^*$ as $t \to \infty$.

