# OpenReview forum: "Convergent Reinforcement Learning with Function Approximation: A Bilevel Optimization Perspective"
_ICLR.cc/2019/Conference_

### Official Review · AnonReviewer2 · 2018-11-03
**Some imprecisions, but interesting new perspective**

**Rating:** 5
**Confidence:** 4

**Review:**

The paper casts the problems of value learning and policy optimization, which can be problematic the non-linear setting, into the bilevel optimization framework. It proposes two novel algorithms with convergence guarantees. Although other works with similar guarantees exist, these algorithms are very appealing for their simplicity. A limited empirical evaluation is provided for the value-based method in Acrobot and Mountain Car and in the Atari games Pong and Breakout for the proposed bilevel Actor Critic.

There are a few missing references to similar, recent work, including Dai et al’s saddle-point algorithm (https://arxiv.org/pdf/1712.10285.pdf). Also, the claim that this is “the first attempt to study the convergence of online reinforcement learning algorithms with nonlinear function approximation” can’t be true (even replacing ‘attempt’ by ‘successfully’, there is e.g. Maei et al.’s nonlinear GTD paper, see below).

Although certainly interesting, the claims relating bilevel optimization and the target network are not completely right. E.g. Equation 3.6 as given is a hard constraint on omega. More explicitly, there are no guarantees that either network is the minimizer of the RHS quantity in 3.6.

The two-timescale algorithm is closer in spirit to the use of a target network, but in DQN and variants the target network is periodically reset, as opposed to what the presented theory would suggest. A different breed of “soft target” networks, which is more closely related to bilevel optimization has been used to stabilize training in DDPG (https://arxiv.org/abs/1509.02971).

There was some confusion for me on the first pass that you define two algorithms called ‘online Q-learning’ and ‘actor-critic’. Neither algorithm is actually that, and they should be renamed accordingly (perhaps ‘bilevel Q-Learning’ and ‘bilevel actor-critic’?). In particular, standard Q-Learning is online; and the actor-critic method does not minimize the Bellman residual (i.e. I believe the RHS of 3.8 is novel within policy-gradient methods).

Once we’re operating on a bounded space with continuous operators, Theorem 4.2 is not altogether surprising – a case of Brouwer’s fixed point theorem, short of the result that theta* = omega*, which is explained in the few lines below the theorem. While I do think Theorem 4.2 is important, it would be good to contrast it to existing results from the GTD family of approaches. Also, requiring that |Q_theta(s,a)| <= Qmax is a significant issue -- effectively this test fails for most commonly used value-based algorithms.

The empirical evaluation lacks any comparison to baselines and serves for little more than as a sanity check of the developed theory. This is probably the biggest weakness of the paper, and is unfortunate given the claim of relevance to e.g. deep RL.



Questions

Throughout, the assumption of the data being sampled on-policy is made without a clear argument as to why. Would the relaxation of this assumption affect the convergence results?

Can the authors provide an intuitive explanation if/why bilevel optimization is necessary?

Can you contrast your work with Maei et al., “Convergent Temporal-Difference Learning with Arbitrary Smooth Function Approximation”?


Suggestions

The discussion surrounding the target network should be improved. In particular, claiming that the DQN target network can be viewed “as the parameter of the upper level optimization subproblem” is a stretch from what is actually shown.

The paper was sometimes hard to follow, in part because the claims are not crisply made. I strongly encourage the authors to more clearly relate their results to existing work, and ensure that the names they use match common usage.

I would have liked to know more about bilevel optimization, what it aims to solve, and the tools used to do it. Instead all I found was very standard two time-scale methods, which was a little disappointing – I don’t think these have been found to work particularly well in practice. This is particularly relevant in the context of e.g. the target network question.

A proper empirical comparison to existing algorithms would significantly improve the quality and relevancy of this work. There are tons of open-source baselines out there, in particular good state of the art implementations. Modifying a standard implementation to optimize its target network along the lines of bilevel optimization should be relatively easy.

Revision:
I thank the authors for their detailed feedback, but still think the work isn't quite ready for publication. After reading the other reviews, I will decrease my score from 6 to 5. Some sticking points/suggestions:
- Some of my concerns remain unanswered. E.g. the actor-critic method 3.8 is driven by the Bellman residual, which is not the same as e.g. the MSPBE used with linear function approximation. There is no harm in proposing variations on existing algorithms, and I'm not sure why the authors are reluctant to do. Also, Brouwer's fixed point theorem, unlike Banach's, does not require a contractive mapping.
- The paper over-claims in a number of places. I highly recommend that the authors make their results more concrete by demonstrating the implications of their method on e.g. linear function approximation. This will also help contrast with Dai et al., etc.

---

> ### Author Response · Authors · 2018-11-26
> **Compare with [Dai et. al.] and [Maei et al.]. Why target network in DQN can be explained using bilevel optimization. Our numerical experiments utilize standard implementations in Deep RL.**
>
> 1. [Dai et al.] We note that [Dai et. al.] is, in reality, a Batch algorithm since their maximization step is assumed to be solved to the global maximum based on data in the experience replay memory. In specific, in Line 7 of their algorithm, they assume that the global maximum of the empirical loss function computed based on the replay memory is achieved. However, since this objective is non-concave, we believe that this condition is strong and can hardly be guaranteed since maximizing a non-concave function is NP-hard. Since this step is solved based on batch data, it is not an online algorithm. Please also refer to Point 2 for Reviewer 1 and Point 2 for Reviewer 2 for detailed discussions on [Dai et. al]
>
> 2. Nonlinear GTD paper. Although the nonlinear GTD paper [Maei et al.] provides a convergence proof for GTD with nonlinear function approximation, they only focus on the problem of policy evaluation. In contrast, we propose a general framework that can incorporate both value function estimation and policy optimization. Our claim is that we first attempt to understand RL algorithms with nonlinear function approximation from a general perspective.
>
> Moreover, using Fenchel duality as in [1] and [2], we could also formulate nonlinear GTD into a saddle point optimization problem, which is a special case of bilevel optimization. Thus, our framework in Section 3.1 also includes nonlinear GTD as a special case.
>
> 3. For (3.6), we assume that for each $\omega$, there exists a global minimizer $\theta(\omega)$ of the least-squares error. Under mild conditions on the function class $\{Q_{\theta} \}$, such a minimizer exists. Moreover, if we could find such an $\omega$ such that $\theta(\omega) = \omega$, then $Q_{\omega}$ is the fixed point of the Bellman operator, which is the optimal Q-function.
>
> Moreover, if the function class $\{Q_{\theta} \}$ is sufficiently large such that it contains the optimal Q-function, then the optimization problem (3.6) has a solution, which is exactly the optimal Q-function.
>
> 4. Our algorithm is more closely related to the ``soft target'' update of the target network, which updates the weights of the target network using a small learning rate $\tau << 1$. In contrast, the Q-network is updated using a constant learning rate, which is much larger than $\tau$. In fact, this corroborates our claim that the target network should be updated in a slower time-scale compared with the Q-network.
>
> Moreover, for the periodical update of the target network, the period is usually set to be a large number such as 5000. This essentially means that we would like to first fix the target network and only update the Q-network, which is exactly solving the lower-level minimization problem in (3.6). Hence, using a bilevel perspective, we provide a theoretical justification of the common practice that the target network is updated at a slower rate compared with Q-network.
>
> 5. The algorithms in our paper are the same as the classical Q-learning and actor-critic algorithms. Thus we call this ```online Q-learning'' and ``actor-critic algorithm''. The goal of our paper is not proposing new algorithms. Instead, our goal is to provide a unified view of the online RL algorithms and provide convergence proofs for these algorithms with nonlinear function approximation. We believe that our work is of interest to deep RL community.

---

> > ### Author Response · Authors · 2018-11-26
> > **(Rebuttal Continued) Compare with [Dai et. al.] and [Maei et al.]. Why target network in DQN can be explained using bilevel optimization. Our numerical experiments utilize standard implementations in Deep RL.**
> >
> >
> > 6. Theorem 4.2 is non-trivial because it concerns the updates of an online algorithm. Moreover, with function approximation, the Brouwer's fixed point theorem no longer holds because we need to consider an operator on the parameter space, which is not contractive. In addition, the boundedness of the value function can be ensured if we restrict the parameters to compact sets in the Euclidean space. This condition is mainly technical. The fact that rewards are bounded implies that the true value functions are bounded.
> >
> > 7. In the experiments, we use neural networks for the value functions and policies and the network structures are the same as standard implementations. Our goal is to verify the observations from our bilevel framework. That is, we verify that the target network in DQN and the policy network in AC should be updated in a slower time-scale. To verify this idea, using standard implementations, we have tried various learning rate configurations on a few standard environments. The numerical results corroborate with our theory.
> >
> > 8. The results of the value function estimation are off-policy. Specifically, $\rho$ is the stationary distribution induced by the behavior policy. Our actor-critic algorithm is on-policy. It is known that there are divergent examples for off-policy TD(0) with nonlinear function approximation. It is our future research that under what conditions can we have convergence guarantees for nonlinear off-policy TD(0).
> >
> > 9. Bilevel optimization for DQN is motivated by the fact that $Q$-network is often updated with the target network fixed. Essentially, this means that we would like to first minimize the TD-error of each fixed target network and then update the weights of the target network by that of the Q-network.
> >
> > The intuition of bilevel optimization for AC can be seen from the policy gradient theorem, where the value function appears. Thus, to estimate the policy gradient, we first need to estimate the value function of the given policy. Thus, essentially, we would like to first solve the policy evaluation problem and then update the policy parameter. Combining these two steps together, we obtain the bilevel optimization problem.
> >
> > 10. We thank the reviewer for the suggestions. We will add detailed discussion on the target network of DQN and also related work.
> > In addition, we note that the experiments in this work follow the standard implementations of DQN and Actor-Critic algorithms. The hyper-parameters are also standard. We only modify the learning rates in order to corroborate with our claims drawn from the two-timescale algorithms.
> > In terms of bilevel optimization, our goal is not to motivate new algorithms from this framework. Instead, we aim to bridge the existing algorithms via this perspective and provide unified convergence analysis. Moreover, using this framework, we give a theoretical justification of the common practice in DQN that using a target network which is updated slowly helps the performance.
> >
> > Reference:
> >
> > [1]. Finite-Sample Analysis of Proximal Gradient TD Algorithms by Liu et al.
> >
> > [2]. SBEED: Convergent Reinforcement Learning with
> > Nonlinear Function Approximation by Dai et. al.

---

### Official Review · AnonReviewer3 · 2018-11-05
**Incremental theoretical result on Q-learning/actor-critic algorithms; Experiments are quite small-scale**

**Rating:** 6
**Confidence:** 4

**Review:**

In this paper the authors studied reinforcement learning algorithms with nonlinear function approximation. By formulating the problems of value function estimation and policy learning as a bilevel optimization problems, the authors proposed
Q-learning and actor-critic algorithms that also contains convergence properties, even when nonlinear function approximations are used.  Similar to the stochastic approximation approach adopted by many previous work such as Borkar (https://core.ac.uk/download/pdf/148488247.pdf), they analyze the convergence properties by drawing connections to stability of a two-timescale ODE. Furthermore they also evaluated the effectiveness of the modified Q-learning/actor-critic algorithms on two toy examples.

In general I find this paper interesting in terms of addressing a long-standing open question of convergence analysis of actor-critic/Q-learning algorithms, when general nonlinear function approximations are used. Through reformulating the problem of value estimation and policy improvement as a bilevel optimization problem, they proposed modifications of Q-learning and actor-critic algorithms, and under certain assumptions they showed that these algorithms converge, which is a non-trivial contribution.

While I appreciate the effort of extending existing analysis of these RL algorithms to general nonlinear function approximation,  I find the result of this paper rather incremental. While convergence results are provided, I am not sure how practical are the assumptions listed in the paper. Correct me if i am wrong, it seems that the assumptions are stated for the sake of proving the theoretical results without much practical justifications (especially Assumption 4.3).  Furthermore how can one ensure that these assumptions hold (for example Assumption 4.3 (i) and (ii), especially on the existence of locally stable equilibrium point) ? Unfortunately I haven't had a chance to go over all the proof details, it seems to me the analysis is built upon two-time scale stochastic approximation theory, which is a standard tool in convergence analysis of actor-critic. Since the contribution of this paper is mostly theoretical, can the authors highlight the novel contribution (such as proof techniques used here that are different than that in standard actor-critic analysis from e.g. https://www.semanticscholar.org/paper/Natural-actor-critic-algorithms-Bhatnagar-Sutton/6a40ffc156aea0c9abbd92294d6b729d2e5d5797)  in the main paper?

My other concern is on the scale of the experiments. While this paper focused on nonlinear function approximation, the examples chosen to evaluate these algorithms are rather small-scale. For example the domains to test Q-learning are standard in RL, and they were previously used to test algorithms with linear function approximation. Can the author compare their results with other existing baselines?

---

> ### Author Response · Authors · 2018-11-26
> **Justification of Assumption 4.3. Our proof requires weaker assumptions than Bhatnagar et. al and the proof requires handling projection, which is not trivial.**
>
>
> 1. Assumption 4.3. The first part of Assumption 4.3 considers the family of value functions and policies. Specifically, we assume that the value functions, as functions of $\theta$, are bounded and have bounded and Lipschitz gradient. This assumption can be satisfied if the parameter $\theta$ lies in a compact set. As a simple example, we could set $V_{\theta}(s) = \sigma( \phi(s)^\top \theta)$, where $\phi(s)$ is a bounded feature mapping, $\sigma$ is the Leaky ReLU activation function. In addition, we also assume that the score function $grad \log \pi_\omega$ is bounded. This assumption is also required in classical convergence proof of actor-critic algorithms. This consider is satisfied if $\pi_{\omega} (s,a) \propto \exp( \phi(s,a)^\top \omega)$ is an energy-based policy with feature mapping $\phi(s,a)$ bounded.
>
> For the second condition, the assumption that the ODE has a local asymptotically stable equilibrium for each $\omega$ essentially means that TD(0) for each policy $\pi_{\omega}$ is convergent. Moreover, the solution is Lipschitz with respect to $\omega$. We admit that this condition is strong since TD(0) with nonlinear function approximation can be divergent. We will clarify this condition in the revised version.
>
>
> 2. In terms of the convergence technique, since we need to handle nonlinear function approximation, we apply projection to the updates to ensure stability. In contrast, since the value functions in [1] are linear, TD(0) is known to converge to the global minimizer of the mean-squared projected Bellman error. Moreover, they assume that both the two ODEs in the faster and slower time-scales have globally asymptotically stable equilibria. Thus, they do not need to apply projection to the policy parameter, which makes their analysis much simpler. However, even with linear function approximation, it seems unnatural to assume that the ODE in slower time-scale has a unique globally asymptotically stable equilibrium since $J(\omega)$ is in general nonconvex in $\omega$. In this work, we provide convergence analysis under much weaker conditions and incorporate nonlinear value functions. Our proof is somewhat more involved than that in [1] due to handling projection.
>
> 3. We have tested the actor-critic algorithm on Atari games, which is large-scale and cannot be solved by algorithms with linear function approximation. Our goal is to verify the idea that the $Q$-network in DQN and the critic in actor-critic algorithm should be updated in the faster timescale.
>
> [1]. Natural Actor–Critic Algorithms by Bhatnagar et. al.

---

> > ### Comment · AnonReviewer3 · 2018-12-05
> > **Thank you for the response**
> >
> > After reading the rebuttals, I appreciate the authors addressed most of the my concerns above. Thus I have adjusted my scores to reflect that. I think the work contains some theoretical contributions as opposed to [1], which is used to prove convergence for many actor-critic algorithms, under certain regular assumptions. It would be great to see this result working on more general conditions. Please address the above concerns in the final version, and also stress the theoretical contributions as compared to existing results.

---

### Official Review · AnonReviewer1 · 2018-11-07
**Interesting work but some of the claims need to be adjusted.**

**Rating:** 6
**Confidence:** 4

**Review:**

This paper interprets the fitted Q-learning, policy evaluation and actor-critic as a bi-level optimization problem. Then, it uses two-timescale stochastic approximation to prove their convergence under nonlinear function approximation. It provides interesting view of the these existing popular reinforcement learning algorithms that is widely used in DRL. However, there are several points to be addressed in the revision, which are mainly in some of its claims.

This paper is mainly a theoretical paper and experiments are carried out on a few simple tasks (Acrobot, MountarinCar, Pong and Breakout). Therefore, it cannot be claimed as “thorough numerical experiments are conducted” as in abstract. This claim should be modified.

Furthermore, it cannot be claimed that this paper is a “first attempt to study the convergence of online reinforcement learning algorithms with nonlinear function approximation in general”. There is a recent work [1], which developed a provably convergent reinforcement learning algorithm with nonlinear function approximation even in the off-policy learning setting.
[1] B. Dai, A. Shaw, L. Li, L. Xiao, N. He, Z. Liu, J. Chen, L. Song, “SBEED Learning: Convergent Control with Nonlinear Function Approximation”, ICML, 2018.

The actor-critic algorithm in the paper uses TD(0) as its policy evaluation algorithm. It is known that the TD(0) algorithm will diverge in nonlinear function approximation and in off-policy learning case. I think the actor-critic algorithm analyzed in the paper is for on-policy learning setting. The authors need to clarify this. Furthermore, the authors may need to comment on how to extend the results to off-policy learning setting.

---

> ### Author Response · Authors · 2018-11-26
> **Our work is different from Dai et. al, which is a batch RL algorithm and involves solving non-concave maximization problems to global maximum.**
>
>
>
> 1. We will add more numerical experiments in the revised version and modify the claim of ``thorough experiments''.
>
> 2. Although [Dai et. al] formulates soft-Q learning as a minimax optimization problem and propose a convergent algorithm, their algorithm is actually not online. The reason is that in the inner maximization step, they need to solve the non-concave maximization to the global maximum. They achieve this step by solving a batch optimization using experience replay and assume that the global maximum can be found by an optimization oracle. Please see Line 7 of Algorithm 1 in [Dai et. al] (https://arxiv.org/pdf/1712.10285.pdf, page 6).
>
> Thus, given the fact that [Dai et. al] is not an online algorithm, we believe that our claim that our work is the ``first attempt to study the convergence of online reinforcement learning algorithms with nonlinear function approximation in general'' is correct.
>
> Moreover, we note that soft Q-learning, which is considered in [Dai et. al], can also be formulated as bilevel optimization by replacing the Bellman operator in (3.7) by the smoothed Bellman operator.
>
> Furthermore, our Algorithm 1 is also an off-policy algorithm, where $\rho$ is the stationary distribution on $(S\times A)$ induced by the behavior policy. Please see the second paragraph on Page 5. We will emphasize that our method is off-policy in the revised version.
>
> We thank the reviewer for listing this related work. We will add detailed discussions of this work in revision. Please also see Point 2 in the response to Reviewer 1 for more discussions of [Dai et. al].
>
> 3. Our actor-critic algorithm is on-policy. We will clarify this in the revised version. For off-policy actor-critic with function approximation, to the best of our knowledge, [1] is the only paper with convergence analysis. The critic update is either GTD(\lambda) or Emphatic-TD(\lambda) and linear function approximation is applied. Their algorithm can also be easily incorporated by the bilevel optimization framework. We will discuss this work in the revised version.
>
> Related work:
> [1]. Convergent Actor-Critic Algorithms Under Off-Policy Training and Function
> Approximation

---

### Official Review · AnonReviewer4 · 2018-11-08
**Interesting theoretical work, but missing key previous literature**

**Rating:** 5
**Confidence:** 3

**Review:**

The authors frame value function estimation and policy learning as bilevel optimization problems, then present a two-timescale stochastic optimization algorithm and convergence results with non-linear function approximators. Finally, they relate the use of target networks in DQN to their two-timescale procedure.

The authors claim that their first contribution is to "unify the problems of value function estimation and policy learning using the framework of bilevel optimization." The bilevel viewpoint has a long history in the RL literature. Are the authors claiming novelty here? If so, can they clarify which parts are novel?

The paper is missing important previous work, SBEED (Dai et al. 2018) which shows (seemingly much stronger) convergence results for a smoothed RL problem. The authors need to compare their approach against SBEED and clearly explain what more they are bringing. Furthermore, the Fenchel trick used in SBEED could also be used to attack the "double sampling" issue here, resulting in a saddle-point problem (which is more specific than the bilevel problem). Does going to the bilevel perspective buy us anything?

=====

In response to the author's comments, I have increased my score.
The practical implications of this theoretical work are unclear. It's nice that it relates to DQN, but it does not provide additional insight into how to improve existing approaches. The authors could significantly strengthen the paper by expanding in this area.

---

> ### Author Response · Authors · 2018-11-26
> **Our bilevel framework seems novel. Dai et al is not an online algorithm and only focus soft-Q learning, which can also be formulated under our framework.**
>
> 1. The bilevel viewpoint. Although the bilevel optimization perspective might be used implicitly in the RL literature, to the best of our knowledge, our work is the first to rigorously bridge the problems of Q-learning and actor-critic under the framework of bilevel optimization. The reviewer seems to believe that there is existing work that has provided such a framework already. If would be great if the reviewer could point us to an example of related work. Moreover, we believe that our formulations of Q-learning and policy learning as constrained bilevel optimization problems are novel.
>
> 2. We will discuss [Dai et. al.] in the revised version. Specifically, they propose a primal-dual formulation for soft Q-learning, which is a value function estimation problem that aims to find the fixed point of the smoothed Bellman operator. This problem can also be formulated as a bilevel optimization problem similar to (3.6).
>
> The differences between our paper and [Dai et. al.] are as follows.
>
> (1). Their problem is formulated as minimax optimization where both the inner maximization and outer minimization problems are neither convex nor concave. They propose a batch algorithm for which, in each iteration, requires solving the inner non-concave maximization problem to its global optima, which can hardly be satisfied in practice.
>
> In contrast, we propose online TD-learning algorithms that are shown to be convergent.
>
> (2). They study only a particular example of value function estimation, which falls in our bilevel optimization framework. In addition to value function estimation, our framework also includes actor-critic, generative adversarial imitation learning, and inverse reinforcement learning.
>
> (3). [Dai et. al] use the Fenchel duality to attack the double sampling issue in soft Q-learning, where the dual function tracks the TD-error. Thus, their dual function essentially is the same as our $Q_{\theta} - T Q_{\omega}$, which implies that their dual function plays a similar role as the target network. Moreover, their Fenchel duality approach cannot be applied to actor-critic. In contrast, our bilevel formulation is more general than their Fenchel duality view.
>
>
> 3. In terms of numerical experiments, our goal is to show that two-time-scale learning rates proposed in Algorithm 1 is essential. The two-time-scale learning rates are motivated by the bilevel formulation. When applied to Q-learning, this implies that the target network should be updated at a slower rate compared with the Q-network. When applied to actor-critic, this implies that the critic should be updated at a faster rate. These observations are further corroborated in the experiments. We believe that these ideas are also useful in the choice of learning rates for practitioners.

---

> > ### Comment · AnonReviewer4 · 2018-11-26
> > **Response**
> >
> > 1. "Connecting Generative Adversarial Networks and Actor-Critic Methods." (Pfau and Vinyals, 2017) for an example.
> >
> > 2. Yes, the saddle point problem is more specific than the bilevel setting. This seems beneficial.
> >
> > 2.1. Are the assumptions in your two-time-scale analysis approach more practical than the analysis in Dai et al.? It would be helpful to discuss why the assumptions you make are likely to hold in practice.
> >
> > 2.2. Okay, these additional applications are interesting.
> >
> > 2.3. Agreed that it plays a similar role.
> >
> > 3. Are the authors claiming that the proposal to use a slower update for the target network is a novel practical contribution, or that their theory motivates an existing practice?

---

> > > ### Author Response · Authors · 2018-11-27
> > > **Compare with related work. Target network**
> > >
> > > We greatly thank the reviewer for reading our response.
> > >
> > > 1. [Pfau et. al.] We thank the reviewer for listing this related work. This paper draws the connection between GAN and actor-critic and reviews the techniques that stabilizes training for these two methods, respectively. They show that both these two problems can be formulated as bilevel optimization and they formulate GAN as a kind of Actor-Critic without states.
> > >
> > > Compared with this work, we focus only on the bilevel formulation of reinforcement learning problems. Our goal is to show that such a formulation unifies both the problem of value function estimation and policy optimization, two of the most important approaches in RL. The bilevel formulation of value function estimation is not covered in [Pfau et. al.].
> > >
> > > Moreover, it is shown in [1] that GAN trained by two-time-scale gradient updates converges to a local Nash equilibrium. Thus, the connection between GAN and AC showed in [Pfau et. al.] in fact corroborates our usage of two-time-scale algorithms for Q-learning and actor-critic. Furthermore, compared with [1] which assumes both the ODEs in the faster and slower time-scales have local asymptotically stable attractors, our assumption is weaker. Our analysis also handles projection of the updates whereas [1] assumes that the gradient updates are always bounded, which we believe might be a strong assumption.
> > >
> > > 2.1. [Dai et. al.] We first note that the algorithm in [Dai et. al.] is a batch RL algorithm whereas our algorithm is online. To see this, please see Line 7 of Algorithm 1 in [Dai et. al] (https://arxiv.org/pdf/1712.10285.pdf, page 6). In Line 7, they assume that the inner maximization problem is replaced by a sample-based optimization where the objective function is computed from the data in the experience replay buffer. Moreover, this maximization problem is non-concave, whose global maximum might be NP-hard to achieve. However, they assume that there exists an optimization oracle that outputs the global maximum, which is denoted by $\omega_{\rho}^j$ in their algorithm.
> > >
> > > Since the global maximum of a non-concave function can be NP-hard to obtain, we believe that this assumption in [Dai et. al.] is quite strong.
> > >
> > > In terms of the assumptions made on the classes of value functions and policy functions, they also require boundedness of these functions. Compared with their assumption, we further require the gradient of the value functions and policies are bounded, which is not significantly stronger and can be satisfied if the parameters are bounded.
> > >
> > > Moreover, the assumptions 4.1(ii) and 4.3(ii) are technical, which essentially ensures that the gradient updates in the faster time-scale converge. Note that there are divergent examples of TD(0) with nonlinear function approximation. We believe that these assumptions are necessary for convergence analysis. Furthermore, these assumptions are much weaker than the assumption of maximizing non-concave functions to the global maximum in [Dai et. al.]
> > >
> > >
> > > 3. Our formulation of bilevel optimization motivates two-time-scale gradient algorithms. When applied to Q-learning, two-time-scale updating rule implies that the target network needs to be updated in a slower timescale. This coincides with the practices in the real world, as shown in the ``baselines'' package ([2]), which is a standard implementation of deep RL algorithms. Thus, when focusing on the target network, our results provide theoretical justifications of the updating rule for the target network used in practice.
> > >
> > > In specific, in the DQN implementation in ``baselines'' for Atari games ([3]), the target network is updated every 1000 steps. Thus, the target network is fixed for a long time and only the Q-network is updated. This is captured in the lower-level optimization problem of our bilevel framework. Furthermore, in practice, some practitioners use soft updates of target network where they update the target network using a learning rate $\tau$ that is much smaller than the learning rate of the $Q$-network. Our framework also justifies such training technique.
> > >
> > >
> > > Reference:
> > >
> > > [1] GANs Trained by a Two Time-Scale Update Rule
> > > Converge to a Local Nash Equilibrium (https://arxiv.org/pdf/1706.08500.pdf)
> > >
> > > [2] OpenAI Baselines: high-quality implementations of reinforcement learning algorithms https://github.com/openai/baselines
> > >
> > > [3] DQN for Atari games in Baselines: https://github.com/openai/baselines/blob/master/baselines/deepq/experiments/train_pong.py

---

> > > > ### Comment · AnonReviewer4 · 2018-12-11
> > > > **RE:**
> > > >
> > > > 1. Ok
> > > >
> > > > 2. Yes the assumptions are stronger, but the results are much stronger as a result, so it's not surprising.
> > > >
> > > > 3. The practical implications of this theoretical work are unclear. It's nice that it provides motivation for current practices, but it does not provide additional insight into how to improve existing approaches. The authors could significantly strengthen the paper by expanding in this area.

---

### Meta-Review · Area_Chair1 · 2018-12-14
**Solid paper, but unclear significance**

**Confidence:** 4
**Recommendation:** Reject

**Metareview:**

The paper gives an bilevel optimization view for several standard RL algorithms, and proves their asymptotic convergence with function approximation under some assumptions.  The analysis is a two-time scale one, and some empirical study is included.

It's a difficult decision to make for this paper.  It clearly has a few things to be liked: (1) the bilevel view seems new in the RL literature (although the view has been implicitly used throughout the literature); (2) the paper is solid and gives rigorous, nontrivial analyses.

On the other hand, reviewers are not convinced it's ready for publication in its current stage:
(1) Technical novelty, in the context of published works: extra challenges needed on top of Borkar; similarity to and differences from Dai et al.; ...
(2) The practical significance is somewhat limited.  Does the analysis provide additional insight into how to improve existing approaches?  How restricted are the assumptions?  Are the online-vs-batch distinction from Dai et al. really important in practice?
(3) What does the paper want to show in the experiments, since no new algorithms are developed?  Some claims are made based on very limited empirical evidence.  It'd be much better to run algorithms on more controlled situations to show, say, the significance of two timescale updates.  Also, as those algorithms are classic Q-learning and actor-critic (quote the authors in responses), how well do the algorithms solve the well-known divergent examples when function approximation is used?
(4) Presentation needs to be improved.  Reviewers pointed out some over claims and imprecise statements.

While the author responses were helpful in clarifying some of the questions, reviewers felt that the remaining questions needed to be addressed and the changes would be large enough that another full review cycle is needed.